# TempME: Towards the Explainability of Temporal Graph Neural Networks via Motif Discovery

**Jialin Chen**
Yale University
jialin.chen@yale.edu

**Rex Ying**
Yale University
rex.ying@yale.edu

## Abstract

Temporal graphs are widely used to model dynamic systems with time-varying interactions. In real-world scenarios, the underlying mechanisms of generating future interactions in dynamic systems are typically governed by a set of recurring substructures within the graph, known as temporal motifs. Despite the success and prevalence of current temporal graph neural networks (TGNN), it remains uncertain which temporal motifs are recognized as the significant indications that trigger a certain prediction from the model, which is a critical challenge for advancing the explainability and trustworthiness of current TGNNs. To address this challenge, we propose a novel approach, called **Temp**oral **M**otifs **E**xplainer (TempME), which uncovers the most pivotal temporal motifs guiding the prediction of TGNNs. Derived from the information bottleneck principle, TempME extracts the most interaction-related motifs while minimizing the amount of contained information to preserve the sparsity and succinctness of the explanation. Events in the explanations generated by TempME are verified to be more spatiotemporally correlated than those of existing approaches, providing more understandable insights. Extensive experiments validate the superiority of TempME, with up to $8.21\%$ increase in terms of explanation accuracy across six real-world datasets and up to $22.96\%$ increase in boosting the prediction Average Precision of current TGNNs.[1]

## 1 Introduction

Temporal Graph Neural Networks (TGNN) are attracting a surge of interest in real-world applications, such as social networks, financial prediction, *etc.* These models exhibit the ability to capture both the topological properties of graphs and the evolving dependencies between interactions over time [1, 2, 3, 4, 5, 6, 7, 8]. Despite their widespread success, these models often lack transparency, functioning as black boxes. The provision of human-intelligible explanations for these models becomes imperative, enabling a better understanding of their decision-making logic and justifying the rationality behind their predictions. Therefore, improving explainability is fundamental in enhancing the trustworthiness of current TGNNs, making them reliable for deployment in real-world scenarios, particularly in high-stakes tasks like fraud detection and healthcare forecasting [9, 10, 11, 12].

The goal of explainability is to discover what patterns in data have been recognized that trigger certain predictions from the model. Explanation approaches on static graph neural networks have been well-studied recently [13, 14, 15, 16, 17, 18, 19]. These methods identify a small subset of important edges or nodes that contribute the most to the model's prediction. However, the success of these methods on static graphs cannot be easily generalized to the field of temporal graphs, due to the complex and volatile nature of dynamic networks [8, 4, 3]. Firstly, there can be duplicate events occurring at the same timestamp and the same position in temporal graphs. The complicated dependencies

---

[1]The code is available at `https://github.com/Graph-and-Geometric-Learning/TempME`

37th Conference on Neural Information Processing Systems (NeurIPS 2023).

between interactions were under-emphasized by existing explanation approaches [20, 21]. Moreover, the important events should be temporally proximate and spatially adjacent to construct a human-intelligible explanation [22]. We refer to explanations that satisfy these requirements as *cohesive* explanations. As illustrated in Figure 1(a), a non-cohesive explanation typically consists of scattered events (highlighted in purple). For instance, event 1 and event 10 in the disjointed explanation are neither temporally proximate nor spatially adjacent to other explanatory events, leading to a sub-optimal explanation and degrading the inspiration that explanations could bring us. There have been some recent attempts at TGNN explainability [21, 23]. Unfortunately, they all face the critical challenge of generating *cohesive* explanations and fall short of providing human-intelligible insights. Moreover, they entail high computational costs, making them impractical for real-world deployment.

To address the aforementioned challenges of temporal explanations, we propose to utilize temporal motifs in the explanation task. Temporal motifs refer to recurring substructures within the graph. Recent studies [24, 25, 26, 27, 28, 29, 30, 31] demonstrate that these temporal motifs are essential factors that control the generative mechanisms of future events in real-world temporal graphs and dynamic systems. For example, preferential attachment (Figure 1(c)) elucidates the influencer effect in e-commerce marketing graphs [32, 33]. Triadic closure (Figure 1(d)) explains the common-friend rules in social networks [34, 6, 1, 25]. Therefore, they are plausible and reliable composition units to explain TGNN predictions. Moreover, the intrinsic self-connectivity of temporal motifs guarantees the *cohesive* property of the generated explanations (Figure 1(b)).

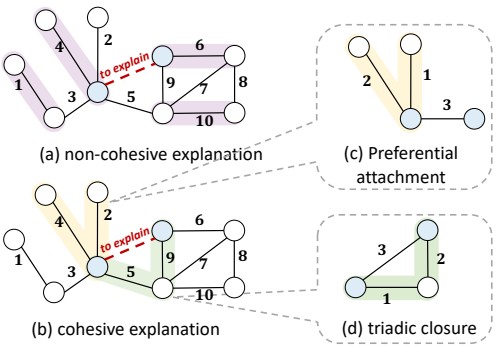

Figure 1: (a) and (b): Non-cohesive explanation and cohesive explanation (highlighted in colors). (c) and (d): Temporal motifs govern the generation of future interactions (numbers denote event orders).

**Proposed work.** In the present work, we propose **TempME**, a novel **Temp**oral **M**otif-based **E**xplainer to identify the most determinant temporal motifs to explain the reasoning logic of temporal GNNs and justify the rationality of the predictions. TempME leverages historical events to train a generative model that captures the underlying distribution of explanatory motifs and thereby improves the explanation efficiency. TempME is theoretically grounded by Information Bottleneck (IB), which finds the best tradeoff between explanation accuracy and compression. To utilize Information Bottleneck in the context of temporal graphs, we incorporate a null model (*i.e.,* a randomized reference) [22, 35, 36] into the model to better measure the information contained in the generated explanations. Thereby, TempME is capable of telling for each motif how the occurrence frequency difference in empirical networks and randomized reference reflects the importance to the model predictions. Different from previous works that only focus on the effect of singular events [23, 21], TempME is the first to bring additional knowledge about the effect of each temporal motif.

We evaluate TempME with three popular TGNN backbones, TGAT [3], TGN [4] and GraphMixer [5]. Extensive experiments demonstrate the superiority and efficiency of TempME in explaining the prediction behavior of these TGNNs and the potential in enhancing the prediction performance of TGNNs, achieving up to 8.21% increase in terms of explanation accuracy across six real-world datasets and up to 22.96% increase in boosting the prediction Average Precision of TGNNs.

The **contributions** of this paper are: (1) We are the first to utilize temporal motifs in the field of explanations for TGNNs to provide more insightful explanations; (2) We further consider the null model in the information bottleneck principle for the temporal explanation task; (3) The discovered temporal motifs not only explain the predictions of different TGNNs but also exhibit ability in enhancing their link prediction performance.

## 2   Related Work

**GNN Explainability**   Explainability methods for Graph Neural Networks can be broadly classified into two categories: non-generative and generative methods. Given an input instance with its

prediction, non-generative methods typically utilize gradients [15, 37], perturbations [38, 39], relevant walks [40], mask optimization [13], surrogate models [41], and Monte Carlo Tree Search (MCTS) [16] to search the explanation subgraph. These methods optimize the explanation one by one during the explanation stage, leading to a longer inference time. On the contrary, generative methods train a generative model across the entire dataset by learning the distribution of the underlying explanatory subgraphs [19, 18, 14, 42, 43, 44, 45], which obtains holistic knowledge of the model behavior over the whole dataset. Compared with static GNN, the explainability of temporal graph neural networks (TGNNs) remain challenging and under-explored. TGNNExplainer [23] is the first explainer tailored for temporal GNNs, which relies on the MCTS algorithm to search for a combination of the explanatory events. Recent work [21] utilizes the probabilistic graphical model to generate explanations for discrete time series on the graph, leaving the continuous-time setting under-explored. However, these methods cannot guarantee cohesive explanations and require significant computation costs. There are also some works that have considered intrinsic interpretation in temporal graphs [26] and seek the self-interpretable models [46, 20]. As ignored by previous works on temporal explanation, we aim for cohesive explanations that are human-understandable and insightful in a generative manner for better efficiency during the explanation stage.

**Network Motifs**    The concept of network motifs is defined as recurring and significant patterns of interconnections [35], which are building blocks for complex networks [47, 24]. Kovanen et al. [22] proposed the first notion of temporal network motifs with edge timestamps, followed by relaxed versions to involve more diverse temporal motifs [48, 24, 49]. Early efforts developed efficient motif discovery algorithms, *e.g.,* MFinder [50], MAVisto [51], Kavosh [36], *etc*. The standard interpretation of the motif counting is presented in terms of a null model, which is a randomized version of the real-world network [35, 52, 22, 53]. Another research line of network motifs focuses on improving network representation learning with local motifs [54, 55, 56]. These approaches emphasize the advantages of incorporating motifs into representation learning, leading to improved performance on downstream tasks. In this work, we constitute the first attempt to involve temporal motifs in the explanation task and target to uncover the decision-making logic of temporal GNNs.

## 3    Preliminaries and Problem Formulation

**Temporal Graph Neural Network**. We treat the temporal graph as a sequence of continuous timestamped events, following the setting in TGNNExplainer [23]. Formally, a temporal graph can be represented as a function of timestamp $t$ by $\mathcal{G}(t) = \{\mathcal{V}(t), \mathcal{E}(t)\}$, where $\mathcal{V}(t)$ and $\mathcal{E}(t)$ denote the set of nodes and events that occur before timestamp $t$. Each element $e_k$ in $\mathcal{E}(t)$ is represented as $e_k = (u_k, v_k, t_k, a_k)$, denoting that node $u_k$ and node $v_k$ have an interaction event at timestamp $t_k < t$ with the event attribution $a_k$. Without loss of generality, we assume that interaction is undirected [5, 1]. Temporal Graph Neural Networks (TGNN) take as input a temporal graph $\mathcal{G}(t)$ and learn a time-aware embedding for each node in $\mathcal{V}(t)$. TGNNs' capability for representation learning on temporal graphs is typically evaluated by their link prediction performance [57, 1, 25], *i.e.,* predicting the future interaction based on historical events. In this work, we also focus on explaining the link prediction behavior of TGNNs, which can be readily extended to node classification tasks.

**Explanation for Temporal Graph Neural Network**. Let $f$ denote a well-trained TGNN (*aka.* base model). To predict whether an interaction event $e$ happens between $u$ and $v$ at timestamp $t$, the base model $f$ leverages the time-aware node representation $x_u(t)$ and $x_v(t)$ to output the logit/probability. An explainer aims at identifying a subset of important historical events from $\mathcal{E}(t)$ that trigger the future interaction prediction made by the base model $f$. The subset of important events is known as an explanation. Formally, let $Y_f[e]$ denote the binary prediction of event $e$ made by base model $f$, the explanation task can be formulated as the following problem that optimizes the mutual information between the explanation and the original model prediciton [13, 23]:

$$\operatorname*{argmax}_{|\mathcal{G}_{\text{exp}}^e| \le K} I(Y_f[e]; \mathcal{G}_{\text{exp}}^e) \quad \Leftrightarrow \quad \operatorname*{argmin}_{|\mathcal{G}_{\text{exp}}^e| \le K} -\sum_{c=0,1} \mathbb{1}(Y_f[e] = c) \log(f(\mathcal{G}_{\text{exp}}^e)[e]) \tag{1}$$

where $I(\cdot, \cdot)$ denotes the mutual information function, $e$ is the interaction event to be explained, $\mathcal{G}_{\text{exp}}^e$ denotes the explanation constructed by important events from $\mathcal{V}(t)$ for $e$. $f(\mathcal{G}_{\text{exp}}^e)[e]$ is the probability output on the event $e$ predicted by the base model $f$. $K$ is the explanation budget on the explanation size (*i.e.,* the number of events in $\mathcal{G}_{\text{exp}}^e$).

# 4 Proposed Method: TempME

A simple optimization of Eq. 1 easily results in disjointed explanations [23]. Therefore, we utilize temporal motifs to ensure that the generated explanations are meaningful and understandable.

The pipeline of TempME is shown in Figure 2. Given a temporal graph and a future prediction between node $u$ and node $v$ to be explained, TempME first samples surrounding temporal motif instances (Sec. 4.1). Then a Motif Encoder creates expressive Motif Embedding for each extracted motif instance, which consists of three main steps: event anonymization, message passing, and graph pooling (Sec. 4.2). Based on Information-bottleneck (IB) principle, TempME characterizes the importance scores of these temporal motifs, under the constraints of both explanation accuracy and information compression (Sec. 4.3). In the explanation stage, succinct and cohesive explanations are constructed by sampling from the Bernoulli distribution controlled by the importance score $p$ for the prediction behavior of the base model.

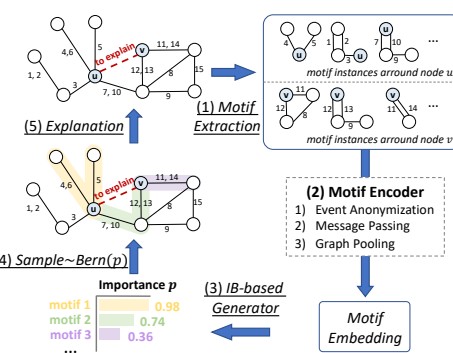

Figure 2: Framework of TempME. Numbers on the edge denote the event order.

## 4.1 Temporal Motif Extraction

We first extract a candidate set of motifs whose importance scores are to be derived. Intuitively, event orders encode temporal causality and correlation. Therefore, we constrain events to reverse over the direction of time in each motif and propose the following Retrospective Temporal Motif.

**Definition 1.** *Given a temporal graph and node $u_0$ at time $t_0$, a sequence of $l$ events, denotes as $I = \{(u_1, v_1, t_1), (u_2, v_2, t_2), \cdots, (u_l, v_l, t_l)\}$ is a n-node, l-length, $\delta$-duration **Retrospective Temporal Motif** of node $u_0$ if the events are reversely time ordered within a $\delta$ duration, i.e., $t_0 > t_1 > t_2 \cdots > t_l$ and $t_0 - t_l \leq \delta$, such that $u_1 = u_0$ and the induced subgraph is connected and contains n nodes.*

Temporal dependencies are typically revealed by the relative order of the event occurrences rather than the absolute time difference. Consequently, we have the following definition of equivalence.

**Definition 2.** *Two temporal motif instances $I_1$ and $I_2$ are **equivalent** if they have the same topology and their events occur in the same order, denoted as $I_1 \simeq I_2$.*

Temporal Motifs are regarded as important building blocks of complex dynamic systems [52, 35, 22, 50]. Due to the large computational complexity in searching high-order temporal motifs, recent works show the great potential of utilizing lower-order temporal motifs, *e.g.,* two-length motifs [52] and three-node motifs [24, 55, 58], as units to analyze large-scale real-world temporal graphs. A collection of temporal motifs with up to 3 nodes and 3 events is shown in Appendix B.

Given a temporal graph with historical events $\mathcal{E}$ and node $u_0$ of interest at time $t_0$, we sample $C$ retrospective temporal motifs with at most $n$ nodes and $l$ events, starting from $u_0$ ($\delta$ is usually set as large for the comprehensiveness of motifs). Alg. 1 shows our Temporal Motif Sampling approach, where $\mathcal{E}(S, t)$ denotes the set of historical events that occur to any node in $S$ before time $t$. At each step, we sample one event from the set of historical events related to the current node set. Alg. 1 adapts Mfinder [50], a motif mining algorithm on static graphs, to the scenario of temporal graphs. We could also assign different sampling probabilities to historical events in Step 3 in Alg. 1 to obtain temporally biased samples. Since the purpose of

---

**Algorithm 1:** Temporal Motif Sampling $(\mathcal{E}, n, l, u_0, t_0, C)$; $l \geq n$

Node set: $S_c \leftarrow \{u_0\}$, for $1 \leq c \leq C$
Event sequence: $I_c \leftarrow ()$, for $1 \leq c \leq C$
**for** $c = 1$ *to* $C$ **do**
    **for** $j = 1$ *to* $l$ **do**
        Sample one event $e_j = (u_j, v_j, t_j)$
        from $\mathcal{E}(S_c, t_{j-1})$
        **if** $|S_c| < n$ **then**
            $S_c = S_c \cup \{u_j, v_j\}$
            $I_c = I_c \parallel e_j$

**return** $\{I_c \mid 1 \leq c \leq C\}$

---

our sampling is to collect a candidate set of expressive temporal motifs for the explanation, we implement uniform sampling in Step 3 for algorithmic efficiency.

**Relation to Previously Proposed Concepts**. Recent works [1, 6] propose to utilize temporal walks and recurrent neural networks (RNN) [59] to aggregate sequential information. Conceptually, temporal walks construct a subset of temporal motif instances in this work. In contrast, temporal motifs capture more graph patterns for a holistic view of the governing rules in dynamic systems. For instance, the motif of preferential attachment (Fig. 1(c)) cannot be represented as temporal walks.

## 4.2 Temporal Motif Embedding

In the present work, we focus on explaining the link prediction of temporal graph neural networks. Given an interaction prediction between node $u$ and node $v$ to be explained, we sample $C$ surrounding temporal motif instances starting from $u$ and $v$, respectively, denoted as $M_u$ and $M_v$. Note the proposed framework is also flexible for explaining other graph-related problems. For instance, to explain the node classification on dynamic graphs, we sample $C$ temporal motif instances around the node of interest. Each temporal motif is represented as $(e_1, e_2, \cdots, e_l)$ with $e_i = (u_i, v_i, t_i)$ satisfying Definition 1. We design a Motif Encoder to learn motif-level representations for each surrounding motif in $M_u$ and $M_v$.

**Event Anonymization**. The anonymization technique is at the core of many sequential feature distillation algorithms [1, 6, 60, 61]. Previous works [6, 1] mainly focus on node anonymization, while temporal motifs are constructed by sequences of temporal events. To bridge this gap, we consider the following event anonymization to adapt to temporal motifs. To maintain the inductiveness, we create structural features to anatomize event identities by counting the appearance at certain positions:

$$h(e_i, u, v)[j] = |\{I \mid I \in M_u \cup M_v, I[j] = (u_i, v_i, t); \forall t\}|, \text{ for } i \in \{1, 2, \cdots, l\}. \quad (2)$$

$h(e_i, u, v)$ (abbreviated as $h(e_i)$ for simplicity) is a $l$-dimensional structural feature of $e_i$ where the $j$-th element denotes the number of interactions between $u_i$ and $v_i$ at the $j$-th sampling position in $M_u \cup M_v$. $h(e_i)$ essentially encodes both spatial and temporal roles of event $e_i$.

**Temporal Motif Encoding**. The extracted temporal motif is essentially a subgraph of the original temporal graph. Instead of using sequential encoders, we utilize local message passing to distill motif-level embedding. Given a motif instance $I$ with node set $\mathcal{V}_I$ and event set $\mathcal{E}_I$, let $X_p$ denote the associated feature of node $p \in \mathcal{V}_I$. $E_{pq} = (a_{pq} \parallel T(t - t_{pq}) \parallel h(e_{pq}))$ denotes the event feature of event $e_{pq} \in \mathcal{E}_I$, where $a_{pq}$ is the associated event attribute and $h(e_{pq})$ refers to the structural feature of event $e_{pq}$ (Eq. 2). Note that the impact of motifs varies depending on the time intervals. For instance, motifs occurring within a single day differ from those occurring within a year. Thus, we need a time encoder $T(\cdot)$ which maps the time interval into $2d$-dimensional vectors via $T(\Delta t) = \sqrt{1/d}[\cos(w_1 \Delta t), \sin(w_1 \Delta t), \cdots, \cos(w_d \Delta t), \sin(w_d \Delta t)]$ with learnable parameters $w_1, \cdots, w_d$ [52, 1]. To derive the motif-level embedding, we initially perform message passing to aggregate neighboring information and then apply the READOUT function to pool node features.

$$\tilde{X}_p = \text{MESSAGEPASSING}(X_p; \{X_q; E_{pq}|q \in \mathcal{N}(p)\}) \text{ and } m_I = \text{READOUT}(\{\tilde{X}_p, p \in \mathcal{V}_I\}) \quad (3)$$

Following Eq. 3, one may use GIN [62] or GAT [63] in MESSAGEPASSING step and simple mean-pooling or learnable adaptive-pooling [64] as READOUT function to further capture powerful motif representations. We refer to Appendix D.4 for more details about the Temporal Motif Encoder.

## 4.3 Information-Bottleneck-based Generator

**Motivation**. A standard analysis for temporal motif distribution is typically associated with the null model, a randomized version of the empirical network [22, 50, 35]. The temporal motif that behaves statistically differently in the occurrence frequency from that of the null model is considered to be structurally significant. Therefore, we assume the information of temporal motifs can be disentangled into interaction-related and interaction-irrelevant ones. The latter is natural result of the null model. Based on this assumption, we resort to the information bottleneck technique to extract compressed components that are the most interaction-related. We refer to Appendix C for theoretical proofs.

**Sampling from Distribution**. Given an explanation query and a motif embedding $m_I$ with $I \in \mathcal{M}$, where $\mathcal{M}$ denotes the set of extracted temporal motifs, we adopt an MLP for mapping $m_I$ to an importance score $p_I \in [0, 1]$, which measures the significance of this temporal motif instance for the explanation query. We sample a mask $\alpha_I \sim \texttt{Bernoulli}(p_I)$ for each temporal motif instance and then apply the masks to screen for a subset of important temporal motif instances via $\mathcal{M}_{\texttt{exp}} = A \odot \mathcal{M}$.

$A$ is the mask vector constructed by $\alpha_I$ for each motif $I$ and $\odot$ denotes element-wise product. The explanation subgraph for the query can thus be induced by all events that occur in $\mathcal{M}_{\exp}$.

To back-propagate the gradients *w.r.t.* the probability $p_I$ during the training stage, we use the Concrete relaxation of the Bernoulli distribution [65] via $\texttt{Bernoulli}(p) \approx \sigma(\frac{1}{\lambda}(\log p - \log(1-p) + \log u - \log(1-u)))$, where $u \sim \texttt{Uniform}(0,1)$, $\lambda$ is a temperature for the Concrete distribution and $\sigma$ is the sigmoid function. In the inference stage, we randomly sample discrete masks from the Bernoulli distribution without relaxation. Then we induce a temporal subgraph with $\mathcal{M}_{\exp}$ as the explanation. One can also rank all temporal motifs by their importance scores and select the Top $K$ important motifs to induce more compact explanations if there is a certain explanation budget in practice.

**Information Bottleneck Objective**. Let $\mathcal{G}_{\exp}^e$ and $\mathcal{G}(e)$ denote the explanation and the computational graph of event $e$ (*i.e.,* historical events that the base model used to predict $e$). To distill the most interaction-related while compressed explanation, the IB objective maximizes mutual information with the target prediction while minimizing mutual information with the original temporal graph:

$$\min -I(\mathcal{G}_{\exp}^e, Y_f[e]) + \beta I(\mathcal{G}_{\exp}^e, \mathcal{G}(e)), \quad s.t. \ |\mathcal{G}_{\exp}^e| \leq K \tag{4}$$

where $Y_f[e]$ refers to the original prediction of event $e$, $\beta$ is the regularization coefficient and $K$ is a constraint on the explanation size. We then adjust Eq. 4 to incorporate temporal motifs.

The first term in Eq. 4 can be estimated with the cross-entropy between the original prediction and the output of base model $f$ given $\mathcal{G}_{\exp}^e$ as Eq. 1, where $\mathcal{G}_{\exp}^e$ is induced by $\mathcal{M}_{\exp}$. Since temporal motifs are essential building blocks of the surrounding subgraph and we have access to the posterior distribution of $\mathcal{M}_{\exp}$ conditioned on $\mathcal{M}$ with importance scores, we propose to formulate the second term in Eq. 4 as the mutual information between the original motif set $\mathcal{M}$ and the selected motif subset $\mathcal{M}_{\exp}$. We utilize a variational approximation $\mathbb{Q}(\mathcal{M}_{\exp})$ to replace its marginal distribution $\mathbb{P}(\mathcal{M}_{\exp})$ and obtain the upper bound of $I(\mathcal{M}, \mathcal{M}_{\exp})$ with Kullback–Leibler divergence:

$$I(\mathcal{M}, \mathcal{M}_{\exp}) \leq \mathbb{E}_{\mathcal{M}} D_{\text{KL}}(\mathbb{P}_\phi(\mathcal{M}_{\exp}|\mathcal{M}); \mathbb{Q}(\mathcal{M}_{\exp})) \tag{5}$$

where $\phi$ involve learnable parameters in Motif Encoder (Eq. 3) and the MLP for importance scores.

**Choice of Prior Distribution**. Different choices of $\mathbb{Q}(\mathcal{M}_{\exp})$ in Eq. 5 may lead to different inductive bias. We consider two practical prior distributions for $\mathbb{Q}(\mathcal{M}_{\exp})$: *uniform* and *empirical*. In the *uniform* setting [42, 66], $\mathbb{Q}(\mathcal{M}_{\exp})$ is the product of Bernoulli distributions with probability $p \in [0, 1]$, that is, each motif shares the same probability $p$ being in the explanation. The KL divergence thus becomes $D_{\text{KL}}(\mathbb{P}_\phi(\mathcal{M}_{\exp}|\mathcal{M}); \mathbb{Q}(\mathcal{M}_{\exp})) = \sum_{I_i \in \mathcal{M}} p_{I_i} \log \frac{p_{I_i}}{p} + (1 - p_{I_i}) \log \frac{1 - p_{I_i}}{1 - p}$. Here $p$ is a hyperparameter that controls both the randomness level in the prior distribution and the prior belief about the explanation volume (*i.e.,* the proportion of motifs that are important for the prediction).

However, *uniform* distribution ignores the effect of the null model, which is a better indication of randomness in the field of temporal graphs. To tackle this challenge, we further propose to leverage the null model to define *empirical* prior distribution for $\mathbb{Q}(\mathcal{M}_{\exp})$. A null model is essentially a randomized version of the empirical network, generated by shuffling or randomizing certain properties while preserving some structural aspects of the original graph. Following prior works on the null model [67, 22], we utilize the common null model in this work, where the event order is randomly shuffled. The null model shares the same degree spectrum and time-shuffled event orders with the input graph [53] (see more details in Appendix D.1). We categorize the motif instances in $\mathcal{M}$ by their equivalence relation defined in Definition B. Let $(U_1, \cdots, U_T)$ denote $T$ equivalence classes of temporal motifs and $(q_1, \cdots, q_T)$ is the sequence of normalized class probabilities occurring in $\mathcal{M}_{\exp}$ with $q_i = \sum_{I_j \in U_i} p_{I_j} / \sum_{I_j \in \mathcal{M}} p_{I_j}$, where $p_{I_j}$ is the importance score of the motif instance $I_j$. Correspondingly, we have $(m_1, \cdots, m_T)$ denoting the sequence of normalized class probabilities in the null model. The prior belief about the average probability of a motif being important for prediction is fixed as $p$. Thus minimizing Eq. 5 is equivalent to the following equation.

$$\min_\phi D_{\text{KL}}(\mathbb{P}_\phi(\mathcal{M}_{\exp}|\mathcal{M}); \mathbb{Q}(\mathcal{M}_{\exp})) \Leftrightarrow \min_\phi (1 - s) \log \frac{1 - s}{1 - p} + s \sum_{i=1}^T q_i \log \frac{sq_i}{pm_i}, \tag{6}$$

where $s$ is computed by $s = \sum_{I_j \in \mathcal{M}} p_{I_j} / |\mathcal{M}|$, which measures the sparsity of the generated explanation. Combing Eq. 1 and Eq. 6 leads to the following overall optimization objective:

$$\min_\phi \mathbb{E}_{e \in \mathcal{E}(t)} \sum_{c=0,1} -\mathbb{1}(Y_f[e] = c) \log(f(\mathcal{G}_{\exp}^e)[e]) + \beta((1 - s) \log \frac{1 - s}{1 - p} + s \sum_{i=1}^T q_i \log \frac{sq_i}{pm_i}). \tag{7}$$

Eq. 7 aims at optimizing the explanation accuracy with the least amount of information. It learns to identify the most interaction-related temporal motifs and push their importance scores close to 1, leading to deterministic existences of certain motifs in the target explanation. Meanwhile, the interaction-irrelevant components are assigned smaller importance scores to balance the trade-off in Eq. 7. TempME shares spirits with perturbation-based explanations [68, 69], where "interpretable components [68]" corresponds to temporal motifs and the "reference" is the null model.

**Complexity**. A brute-force implementation of the sampling algorithm (Alg. 1) has the time complexity $\mathcal{O}(Cl)$. Following Liu et al. [52], we create a $2l$-digit to represent a temporal motif with $l$ events, where each pair of digits is an event between the node represented by the first digit and the node represented by the second digit. We utilize these $2l$-digits to classify the temporal motifs by their equivalence relations, thus resulting in a complexity of $\mathcal{O}(C)$. An acceleration strategy with the tree-structured sampling and detailed complexity analysis are given in Appendix D.3.

## 5 Experiments

### 5.1 Experimental Setups

**Dataset**. We evaluate the effectiveness of TempME on six real-world temporal graph datasets, Wikipedia, Reddit, Enron, UCI, Can.Parl., and US Legis [70, 71, 72] that cover a wide range of domains. Wikipedia and Reddit are bipartite networks with rich interaction attributes. Enron and UCI are social networks without any interaction attributes. Can.Parl. and US Legis are two political networks with a single attribute. Detailed dataset statistics are given in Appendix E.1.

**Base Model**. The proposed TempME can be employed to explain any temporal graph neural network (TGNN) that augments local message passing. We adopt three state-of-the-art temporal graph neural networks as the base model: TGAT [3], TGN [4], and GraphMixer [5]. TGN and GraphMixer achieve high performance with only one layer due to their powerful expressivity or memory module. TGAT typically contains 2-3 layers to achieve the best performance. Following previous training setting [23, 1, 6], we randomly sample an equal amount of negative links and consider event prediction as a binary classification problem. All models are trained in an inductive setting [6, 1].

**Baselines**. Assuming the base model contains $L$ layers and we aim at explaining the prediction on the event $e$, we first extract $L$-hop neighboring historical events as the computational graph $\mathcal{G}(e)$. For baselines, we first compare with two self-interpretable techniques, Attention (ATTN) and Gradient-based Explanation (Grad-CAM [37]). For ATTN, we extract the attention weights in TGAT and TGN and take the average across heads and layers as the importance scores for events. For Grad-CAM, we calculate the gradient of the loss function *w.r.t.* event features and take the norms as the importance scores. Explanation $\mathcal{G}_{\text{exp}}^e$ is generated by ranking events in $\mathcal{G}(e)$ and selecting a subset of explanatory events with the highest importance scores. We further compare with learning-based approaches, GNNExplainer [13], PGExplainer [14] and TGNNExplainer [23], following the baseline setting in prior work [23]. The former two are proposed to explain static GNNs while TGNNExplainer is a current state-of-the-art model specifically designed for temporal GNNs.

**Configuration**. Standard fixed splits [73, 72] are applied on each datasets. Following previous studies on network motifs [52, 22, 35, 56], we have empirically found that temporal motifs with at most 3 nodes and 3 events are sufficiently expressive for the explanation task (Fig. 4). We use GINE [74], a modified version of GIN [62] that incorporates edge features in the aggregation function, as the MESSAGEPASSING function and mean-pooling as the READOUT function by default.

### 5.2 Explanation Performance

**Evaluation Metrics**. To evaluate the explanation performance, we report Fidelity and Sparsity following TGNNExplainer [23]. Let $\mathcal{G}_{\text{exp}}^e$ and $\mathcal{G}$ denote the explanation for event $e$ and the original temporal graph, respectively. Fidelity measures how valid and faithful the explanations are to the model's original prediction. If the original prediction is positive, then an explanation leading to an increase in the model's prediction logit is considered to be more faithful and valid and vice versa. Fidelity is defined as $\text{Fid}(\mathcal{G}, \mathcal{G}_{\text{exp}}^e) = \mathbb{1}(Y_f[e] = 1)(f(\mathcal{G}_{\text{exp}}^e)[e] - f(\mathcal{G})[e]) + \mathbb{1}(Y_f[e] = 0)(f(\mathcal{G})[e] - f(\mathcal{G}_{\text{exp}}^e)[e])$. Sparsity is defined as $\text{Sparsity} = |\mathcal{G}_{\text{exp}}^e|/|\mathcal{G}(e)|$, where $\mathcal{G}(e)$ denotes the computational graph of event $e$. An ideal explanation should be compact and succinct, therefore,

Table 1: ACC-AUC of TempME and baselines over six datasets and three base models. The AUC values are computed over 16 sparsity levels between 0 and 0.3 at the interval of 0.02. The best result is in **bold** and second best is underlined.

|  |  | Wikipedia | Reddit | UCI | Enron | USLegis | Can.Parl. |
|---|---|---|---|---|---|---|---|
| **TGAT** | Random | 70.91±1.03 | 81.97±0.92 | 54.51±0.52 | 48.94±1.28 | 54.24±1.34 | 51.66±2.26 |
|  | ATTN | 77.31±0.01 | 86.80±0.01 | 27.25±0.01 | 68.28±0.01 | 62.24±0.00 | 79.92±0.01 |
|  | Grad-CAM | 83.11±0.01 | 90.29±0.01 | 26.06±0.01 | 19.93±0.01 | 78.98±0.01 | 50.42±0.01 |
|  | GNNExplainer | 84.34±0.16 | 89.44±0.56 | 62.38±0.46 | 77.82±0.88 | 89.42±0.50 | 80.59±0.58 |
|  | PGExplainer | 84.26±0.78 | 92.31±0.92 | 59.47±1.68 | 62.37±3.82 | 91.42±0.94 | 75.92±1.12 |
|  | TGNNExplainer | 85.74±0.56 | 95.73±0.36 | 68.26±2.62 | **82.02±1.94** | 90.37±0.84 | 80.67±1.49 |
|  | **TempME** | **85.81±0.53** | **96.69±0.38** | **76.47±0.80** | 81.85±0.26 | **96.10±0.20** | **84.48±0.97** |
| **TGN** | Random | 91.90±1.42 | 91.42±1.94 | 87.15±2.23 | 82.72±2.24 | 72.31±2.64 | 76.43±1.65 |
|  | ATTN | 93.28±0.01 | 93.81±0.01 | 83.24±0.01 | 83.57±0.01 | 75.62±0.01 | 79.38±0.01 |
|  | Grad-CAM | 93.46±0.01 | 92.60±0.01 | 87.51±0.01 | 81.12±0.01 | 81.46±0.01 | 77.19±0.01 |
|  | GNNExplainer | 95.62±0.53 | 95.50±0.35 | 94.68±0.42 | 88.61±0.50 | 82.91±0.46 | 83.32±0.64 |
|  | PGExplainer | 94.28±0.84 | 94.42±0.36 | 92.39±0.85 | 88.34±1.24 | 90.62±0.75 | 88.46±1.42 |
|  | TGNNExplainer | 93.51±0.98 | 96.21±0.47 | 94.24±0.52 | 90.32±0.82 | 90.40±0.83 | 84.70±1.19 |
|  | **TempME** | **95.80±0.42** | **98.66±0.80** | **96.34±0.30** | **92.64±0.27** | **94.37±0.88** | **90.63±0.72** |
| **GraphMixer** | Random | 77.31±2.37 | 85.08±0.72 | 53.56±1.27 | 64.07±0.86 | 85.54±0.93 | 87.79±0.51 |
|  | Grad-CAM | 76.63±0.01 | 84.44±0.41 | 82.64±0.01 | 72.50±0.01 | 88.98±0.01 | 85.80±0.01 |
|  | GNNExplainer | 89.21±0.63 | 95.10±0.36 | 61.02±0.37 | 74.23±0.13 | 89.67±0.35 | 92.28±0.10 |
|  | PGExplainer | 85.19±1.24 | 92.46±0.42 | 63.76±1.06 | 75.39±0.43 | 92.37±0.10 | 90.63±0.32 |
|  | TGNNExplainer | 87.69±0.86 | **95.82±0.73** | 80.47±0.87 | **81.87±0.45** | 93.04±0.45 | 93.78±0.74 |
|  | **TempME** | **90.15±0.30** | 95.05±0.19 | **87.06±0.12** | 79.69±0.33 | **95.00±0.16** | **95.98±0.21** |

higher fidelity with lower Sparsity denotes a better explanation performance. Besides, we further adopt the `ACC-AUC` metric, which is the AUC value of the proportion of generated explanations that have the same predicted label by the base model over sparsity levels from 0 to 0.3.

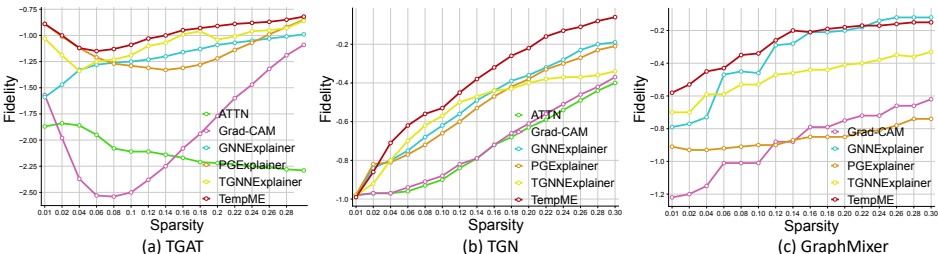

Figure 3: Fidelity-Sparsity Curves on Wikipedia dataset with different base models

**Results**. Table 1 shows the explanation performance of TempME and other baselines *w.r.t.* ACC-AUC. TempME outperforms baselines on different datasets and base models in general. Notably, TempME achieves state-of-the-art performance in explaining TGN with strong ACC-AUC results ($\geq 90\%$) over all six datasets. Specifically, the effectiveness of TempME is consistent across datasets with and without attributes, whereas the performance of baseline models exhibits considerable variation. For example, ATTN and Grad-CAM work well on datasets with rich attributes, *e.g.,* Wikipedia and Reddit, while may yield poor performances on unattributed datasets. Therefore, events with large gradients or attention values are not sufficient to explain the decision-making logic of the base model.

Figure 3 demonstrates the Fidelity-Sparsity curves of TempME and compared baselines on Wikipedia with different base models. From Figure 3, we observe that TempME surpasses the baselines in terms of explanation fidelity, especially with a low sparsity level. In addition, it reveals that the optimal sparsity level varies among different base models. For TGAT, increasing sparsity initially diminishes and later enhances the general fidelity. Conversely, for TGN and GraphMixer, increasing sparsity consistently improves fidelity. These findings indicate that TGAT gives priority to a narrow subset (*e.g.,* 1%) of historical events, while TGN and GraphMixer rely on a wider range of historical events.

**Cohesiveness**. To evaluate the cohesive level of the explanations, we propose the following metric:

$$\text{Cohesiveness} = \frac{1}{|\mathcal{G}^e_{\text{exp}}|^2 - |\mathcal{G}^e_{\text{exp}}|} \sum_{e_i \in \mathcal{G}^e_{\text{exp}}} \sum_{e_j \in \mathcal{G}^e_{\text{exp}}; e_i \neq e_j} \cos(\frac{|t_i - t_j|}{\Delta T}) \mathbb{1}(e_i \sim e_j), \qquad (8)$$

where $\Delta T$ means the time duration in the computational graph $\mathcal{G}(e)$, $\mathbb{1}(e_i \sim e_j)$ indicates whether $e_i$ is spatially adjacent to $e_j$. Meanwhile, temporally proximate event pairs are assigned with larger weights of $\cos(|t_i - t_j|/\Delta T)$. A higher level of cohesiveness indicates a more cohesive explanation. From Table 2, we observe that ATTN and Grad-CAM excel in generating cohesive explanations compared to learning-based explainers, *e.g.,* GNNExplainer, TGNNExplainer. However, TempME still surpasses all baselines and achieves the highest cohesiveness levels, primarily due to its ability to extract and utilize self-connected motifs, allowing it to generate explanations that are both coherent and cohesive.

**Efficiency Evaluation**. We empirically investigate the efficiency of TempME in terms of the inference time for generating one explanation and report the results for Wikipedia and Reddit on TGAT in Table 3, where the averages are calculated across all test events. GNNExplainer and TGNNExplainer optimize explanations individually for each instance, making them less efficient. Notably, TGNNExplainer is particularly time-consuming due to its reliance on the MCTS algorithm. In contrast, TempME trains a generative model using historical events, which allows for generalization to future unseen events. As a result, TempME demonstrates high efficiency and fast inference.

Table 2: Cohesiveness evaluation on Reddit and UCI with TGAT.

|  | Reddit | UCI |
|---|---|---|
| ATTN | 0.0502 | 0.0708 |
| Grad-CAM | 0.0422 | 0.0722 |
| GNNExplainer | 0.0270 | 0.0337 |
| PGExplainer | 0.0233 | 0.0332 |
| TGNNExplainer | 0.0397 | 0.0538 |
| **TempME** | **0.0574** | **0.0749** |

Table 3: Inference time (seconds) of one explanation for TGAT

|  | Wikipedia | Reddit |
|---|---|---|
| Random | 0.02±0.06 | 0.03±0.09 |
| ATTN | 0.02±0.00 | 0.04±0.00 |
| Grad-CAM | 0.03±0.00 | 0.04±0.00 |
| GNNExplainer | 8.24±0.26 | 10.44±0.67 |
| PGExplainer | 0.08±0.01 | 0.08±0.01 |
| TGNNExplainer | 26.87±3.71 | 83.70±16.24 |
| **TempME** | 0.13±0.02 | 0.15±0.02 |

Table 4: Link prediction results (Average Precision) of base models with Motif Embedding (ME)

|  | UCI | Enron | USLegis | Can.Parl. |
|---|---|---|---|---|
| TGAT | 76.28 | 65.68 | 72.35 | 65.18 |
| TGAT+ME | **83.65**$^{(\uparrow 7.37)}$ | **68.37** $^{(\uparrow 2.69)}$ | **95.31** $^{(\uparrow 22.96)}$ | **76.35**$^{(\uparrow 11.17)}$ |
| TGN | 75.82 | **76.40** | 77.28 | 64.23 |
| TGN+ME | **77.46**$^{(\uparrow 1.64)}$ | 75.62$^{(\downarrow 0.78)}$ | **83.90**$^{(\uparrow 6.62)}$ | **79.46**$^{(\uparrow 15.23)}$ |
| GraphMixer | 89.13 | 69.42 | 66.71 | 76.98 |
| GraphMixer+ME | **90.11**$^{(\uparrow 0.98)}$ | **70.13**$^{(\uparrow 0.71)}$ | **81.42**$^{(\uparrow 14.71)}$ | **79.33**$^{(\uparrow 2.35)}$ |

**Motif-enhanced Link Prediction**. The extracted motifs can not only be used to generate explanations but also boost the performance of TGNNs. Let $m_I$ denote the motif embedding generated by the Temporal Motif Encoder (Eq. 3) and $\mathcal{M}$ is the temporal motif set around the node of interest. We aggregate all these motif embeddings using $\sum_{I \in \mathcal{M}} m_I/|\mathcal{M}|$ and concatenate it with the node representations before the final MLP layer in the base model. The performance of base models on link prediction with and without Motif Embeddings (ME) is shown in Table 4. Motif Embedding provides augmenting information for link prediction and generally improves the performance of base models. Notably, TGAT achieves a substantial boost, with an Average Precision of $95.31\%$ on USLegis, surpassing the performance of state-of-the-art models on USLegis [72, 73]. More results are given in Appendix E.3.

**Ablation Studies**. We analyze the hyperparameter sensitivity and the effect of prior distributions used in TempME, including the number of temporal motifs $C$, the number of events in the motifs $l$, and the prior belief about the explanation volume $p$. The results are illustrated in Figure 4.

Firstly, when using smaller motifs (*e.g.*, $l = 2$), TempME achieves comparable explanation accuracy when a sufficient number of motifs are sampled. However, the accuracy plateaus with fewer temporal motifs when $l = 3$ or $l = 4$. Unfortunately, there are only three equivalence classes for temporal motifs with only two events, limiting the diversity of perspectives in explanations. Following previous analysis on temporal motifs [52, 22,

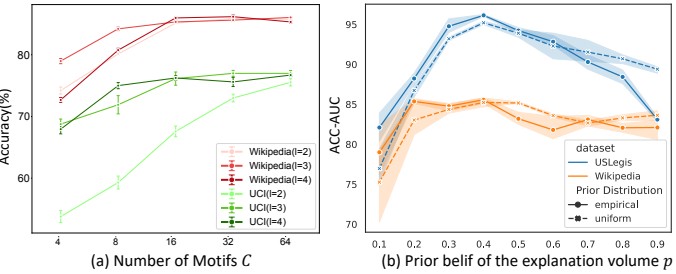

Figure 4: (a) Hyperparameter sensitivity of number of temporal motifs $C$ and motif length $l$. (b) Comparison between uniform and empirical prior distribution in terms of ACC-AUC over sparsity levels from 0 to 0.3.

24], we suggest considering temporal motifs with up to 3 events in the explanation task for the sake of algorithmic efficiency. Secondly, TempME achieves the highest ACC-AUC when the prior belief of the explanation volume is in the range of $[0.3, 0.5]$. Notably, TempME performs better with *empirical* prior distribution when $p$ is relatively small, resulting in sparser and more compact explanations. This improvement can be attributed to the incorporation of the null model, which highlights temporal motifs that differ significantly in frequency from the null model. Figure 4 (b) verifies the rationality and effectiveness of the *empirical* prior distribution in TempME. Additional insight into the role of the null model in explanation generation can be found in the explanation visualizations in Appendix E.5.

It is worth noting that when $p$ is close to 1, *uniform* prior distribution leads to deterministic existences of all temporal motifs while *empirical* prior distribution pushes the generated explanations towards the null model, which forms the reason for the ACC-AUC difference of *empirical* and *uniform* as $p$ approaches 1.

Table 5: Ablation studies on the main components of TempME in terms of the explanation ACC-AUC on Wikipedia.

|  | TGAT | TGN | GraphMixer |
|---|---|---|---|
| **TempME** | **85.81±0.53** | **95.80±0.42** | **90.15±0.30** |
| *Temporal Motif Encoder* | | | |
| w/ GCN | 83.26±0.70 | 94.62±0.34 | 88.62±0.95 |
| w/ GAT | 84.37±0.63 | 95.46±0.73 | 88.59±0.84 |
| w/ Adaptive Pooling | 85.24±0.46 | 95.73±0.27 | **90.15±0.27** |
| w/o Event Anonymization | 81.47±1.14 | 93.77±0.52 | 88.32±0.93 |
| w/o Time Encoding | 79.42±0.82 | 92.64±0.70 | 87.90±0.86 |
| *Loss function* | | | |
| w/ uniform | 85.60±0.75 | 94.60±0.38 | 87.46±0.71 |

We further conduct ablation studies on the main components of TempME. We report the explanation ACC-AUC on Wikipedia in Table 5. Specifically, we first replace the GINE convolution with GCN and GAT and replace the mean-pooling with adaptive pooling in the Temporal Motif Encoder. Then we iteratively remove event anonymization and time encoding in the creation of event features before they are fed into the Temporal Motif Encoder (Eq. 3). Results in Table 5 demonstrate that all the above variants lead to performance degradation. Moreover, the Time Encoding results in a more severe performance drop across three base models. We further evaluate the effectiveness of *empirical* prior distribution by comparing it with *uniform* prior distribution. In both prior distributions, the prior belief on the explanation size $p$ is set to $0.3$. We report the best results in Table 5. We can observe that the *empirical* prior distribution gains a performance boost across three base models, demonstrating the importance of the null model in identifying the most significant motifs.

# 6 Conclusion and Broader Impacts

In this work, we present TempME, a novel explanation framework for temporal graph neural networks. Utilizing the power tool of temporal motifs and the information bottleneck principle, TempME is capable of identifying the historical events that are the most contributing to the predictions made by TGNNs. The success of TempME bridges the gap in explainability research on temporal GNNs and points out worth-exploring directions for future research. For instance, TempME can be deployed to analyze the predictive behavior of different models, screen effective models that can capture important patterns, and online services to improve the reliability of temporal predictions.

By enabling the generation of explainable predictions and insights, temporal GNNs can enhance decision-making processes in critical domains such as healthcare, finance, and social networks. Improved interpretability can foster trust and accountability, making temporal GNNs more accessible to end-users and policymakers. However, it is crucial to ensure that the explanations provided by the models are fair, unbiased, and transparent. Moreover, ethical considerations, such as privacy preservation, should be addressed to protect individuals' sensitive information during the analysis of temporal graphs.

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

# A Notation Table

The main notations used throughout this paper are summarized in Table 6.

Table 6: Summary of the notations

| Notation | Description |
|---|---|
| $\mathcal{G}(t)$ | Continuous-time dynamic graphs |
| $\mathcal{V}(t)$ | Set of nodes that occur before timestamp $t$ |
| $\mathcal{E}(t)$ | Set of events (interactions) that occur before timestamp $t$ |
| $e_k = (u_k, v_k, t_k, a_k)$ | Interaction event $e_k$ between node $u_k$ and $v_k$ at time $t_k$ with attribute $a_k$ |
| $f$ | Temporal Graph Neural Network to be explained (base model) |
| $Y_f[e]$ | Binary prediction of event $e$ made by the base model $f$ |
| $\mathcal{G}(e)$ | Computational graph of event $e$ |
| $\mathcal{G}_{\text{exp}}^e$ | Explanation graph for the prediction of event $e$ |
| $I(\cdot, \cdot)$ | Mutual information function |
| $f(\cdot)[e]$ | Probability output of the model $f$ on the event $e$ |
| $K$ | Explanation budget on the size |
| $I$ | A temporal motif instance |
| $n$ | Maximum number of nodes in temporal motif instances |
| $l$ | Number of events in each temporal motif instance |
| $h(e)$ | $l$-dimensional structural feature of the event $e$ |
| $M_u$ | Set of temporal motif instances starting from the node $u$ |
| $C$ | Number of temporal motif instances sampled for each node of interest |
| $X_p$ | Associated feature of node $p$ |
| $a_{pq}$ | Associated feature of event $e_{pq}$ that happens between node $p$ and node $q$ |
| $T(\cdot)$ | Time encoder |
| $m_I$ | Motif-level embedding of $I$ |
| $p_I \in [0, 1]$ | Importance score of $I$ |
| $\alpha_I \sim \texttt{Bernoulli}(p_I)$ | Mask for $I$ sampled from $\texttt{Bernoulli}(p_I)$ |
| $\mathcal{M}$ | Set of the extracted temporal motifs |
| $\mathcal{M}_{\text{exp}}$ | Set of explanatory temporal motifs |
| $\mathbb{P}_\phi(\mathcal{M}_{\text{exp}}|\mathcal{M})$ | Posterior distribution of $\mathcal{M}_{\text{exp}}$ given $\mathcal{M}$ with learnable parameters $\phi$ |
| $\mathbb{Q}(\mathcal{M}_{\text{exp}})$ | Prior distribution of $\mathcal{M}_{\text{exp}}$ |
| $p$ | Prior belief about the explanation volume |

# B Temporal Motif

Given an explanation query of the future link prediction between node $i$ and node $j$ at time $t_0$, we consider the temporal motifs around node $i$ and node $j$ to explain which motifs contribute to the model's prediction. We first extract two sets of temporal motifs starting from node $i$ and from node $j$ at time $t_0$, respectively. Since we consider the effect of historical events, we constrain events to reverse over time direction in each temporal motif.

**Definition.** *Given a temporal graph and node $u_0$ at time $t_0$, a sequence of $l$ events, denotes as $I = \{(u_1, v_1, t_1), (u_2, v_2, t_2), \cdots, (u_l, v_l, t_l)\}$ is a $n$-node, $l$-length, $\delta$-duration **Retrospective Temporal Motif** of node $u_0$ if the events are reversely time ordered within a $\delta$ duration, i.e., $t_0 > t_1 > t_2 \cdots > t_l$ and $t_0 - t_l \leq \delta$, such that $u_1 = u_0$ and the induced subgraph is connected and contains $n$ nodes.*

There can be a collection of event sequences that satisfy the above definition. Intuitively, two motif instances are equivalent if the order of the events is the same, despite the absolute time difference.

**Definition.** *Two temporal motif instances $I_1$ and $I_2$ are **equivalent** if they have the same topology and their events occur in the same order, denoted as $I_1 \simeq I_2$.*

We use $\{0, 1, \cdots, n-1\}$ to denote the nodes in the motif and use $l$ digit pairs to construct a $2l$-digit to represent each temporal motif with $l$ events. Each pair of digits denotes an event between the node represented by the first digit and the other node represented by the second digit. The first digit pair

is always 01, denoting that the first event occurred between node 0 and node 1. The sequence of digit pairs and the digit number of each node follow the chronological order in each temporal motif. Consider the undirected setting, the examples of temporal motifs with at most 3 nodes and 3 events and their associated digital representations are shown in Figure 5 (a).

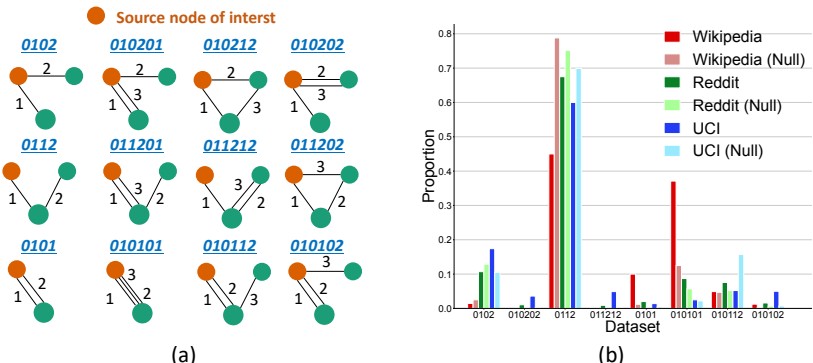

(a)                                    (b)

Figure 5: (a) Visualization of temporal motifs with up to 3 nodes and 3 events and the associated $2l$-digits. (b) The proportion of sampled temporal motif equivalence classes in Wikipedia, Reddit, UCI and their corresponding null models (only a subset of the most frequent motifs are displayed)

## C    Theoretical Proof

The information bottleneck is a technique to find the best tradeoff between accuracy and compression. Given a temporal graph $\mathcal{G}$ and a prediction $Y_f[e]$ over the future event $e$ made by model $f$. The goal of the explanation is to extract a compressed but explanatory subgraph $\mathcal{G}_{\text{exp}}^e$ from the original graph $\mathcal{G}$, such that $\mathcal{G}_{\text{exp}}^e$ plays a pivotal role in leading to the target prediction $Y_f[e]$. It can be formulated as an information bottleneck problem as follows.

$$\min -I(\mathcal{G}_{\text{exp}}^e, Y_f[e]) + \beta I(\mathcal{G}_{\text{exp}}^e, \mathcal{G}(e)), \quad s.t. \ |\mathcal{G}_{\text{exp}}^e| \leq K, \tag{9}$$

where $\mathcal{G}(e)$ denotes the computational graph of event $e$, and $K$ is a constraint on the explanation size (*i.e.,* number of historical events selected into $\mathcal{G}_{\text{exp}}^e$).

### C.1    Accuracy Term: Cross Entropy

The first term in Eq. 9 can be approximated by the cross entropy between the model's prediction given $\mathcal{G}_{\text{exp}}^e$ and the target prediction $Y_f[e]$.

$$\min -I(\mathcal{G}_{\text{exp}}^e, Y_f[e]) = \min H(Y_f[e] \mid \mathcal{G}_{\text{exp}}^e) - H(Y_f[e])$$
$$\Leftrightarrow \min H(Y_f[e] \mid \mathcal{G}_{\text{exp}}^e) = \min -\sum_{c=0,1} -\mathbb{1}(Y_f[e] = c) \log(f(\mathcal{G}_{\text{exp}}^e)[e]) \tag{10}$$

where $H(\cdot)$ is the entropy function and $H(Y_f[e])$ is constant during the explanation stage.

### C.2    Variational Bounds for Information Bottleneck

Let $M_u$ and $M_v$ denote the sets of sampled temporal motif instances surrounding the node $u$ and node $v$. We propose to formulate the second term in Eq. 9 as the mutual information between $\mathcal{M}$ and $\mathcal{M}_{\text{exp}}$, where $\mathcal{M} = M_u \cup M_v$ denotes the set of all extracted motifs and $\mathcal{M}_{\text{exp}}$ is the set of explanatory temporal motifs, since they are the essential building blocks of $\mathcal{G}(e)$ and $\mathcal{G}_{\text{exp}}$. We introduce $\mathbb{Q}(\mathcal{M}_{\text{exp}})$ as a variational approximation for the marginal distribution $\mathbb{P}(\mathcal{M}_{\text{exp}})$ and derive

the variational bounds for the second term in Eq. 9.

$$
\begin{aligned}
I(\mathcal{M}; \mathcal{M}_{\text{exp}}) &= \mathbb{E}_{\mathbb{P}(\mathcal{M},\mathcal{M}_{\text{exp}})}\left[\log \frac{\mathbb{P}(\mathcal{M}_{\text{exp}} \mid \mathcal{M})}{\mathbb{P}(\mathcal{M}_{\text{exp}})}\right] \\
&= \mathbb{E}_{\mathbb{P}(\mathcal{M},\mathcal{M}_{\text{exp}})}\left[\log \frac{\mathbb{P}(\mathcal{M}_{\text{exp}} \mid \mathcal{M})\mathbb{Q}(\mathcal{M}_{\text{exp}})}{\mathbb{Q}(\mathcal{M}_{\text{exp}})\mathbb{P}(\mathcal{M}_{\text{exp}})}\right] \\
&= \mathbb{E}_{\mathbb{P}(\mathcal{M},\mathcal{M}_{\text{exp}})}\left[\log \frac{\mathbb{P}(\mathcal{M}_{\text{exp}} \mid \mathcal{M})}{\mathbb{Q}(\mathcal{M}_{\text{exp}})}\right] - D_{\text{KL}}(\mathbb{P}(\mathcal{M}_{\text{exp}}); \mathbb{Q}(\mathcal{M}_{\text{exp}})) \\
&\leq \mathbb{E}_{\mathbb{P}(\mathcal{M})}[D_{\text{KL}}(\mathbb{P}(\mathcal{M}_{\text{exp}} \mid \mathcal{M}); \mathbb{Q}(\mathcal{M}_{\text{exp}}))].
\end{aligned}
\tag{11}
$$

Let $p_{I_j}$ denote the importance score of the motif instance $I_j \in \mathcal{M}$, which measures the probability of motif instance $I_j$ being sampled as explanatory temporal motifs in $\mathcal{M}_{\text{exp}}$. On average, the proportion of temporal motif instances being selected into $\mathcal{M}_{\text{exp}}$ is $s = \frac{\sum_{I_j \in \mathcal{M}} p_{I_j}}{|\mathcal{M}|}$. Let $\{U_1, \cdots, U_T\}$ denote $T$ equivalence classes that occur in $\mathcal{M}$. Thus, $q_i = \frac{\sum_{I_j \in U_i} p_{I_j}}{\sum_{I_j \in \mathcal{M}} p_{I_j}}$ denotes the proportion of the temporal motifs that belong to equivalence class $U_i$ in $\mathcal{M}_{\text{exp}}$. The prior belief about the average probability of a motif being explanatory is $p$. Assume that in the null model, the proportion of the temporal motifs belonging to $U_j$ is $m_j$. Then we have

$$
\begin{aligned}
I(\mathcal{M}; \mathcal{M}_{\text{exp}}) &\leq \mathbb{E}_{\mathbb{P}(\mathcal{M})}[D_{\text{KL}}(\mathbb{P}(\mathcal{M}_{\text{exp}} \mid \mathcal{M}); \mathbb{Q}(\mathcal{M}_{\text{exp}}))] \\
&= \mathbb{E}_{\mathbb{P}(\mathcal{M})} \sum_{U_j, j=1,\cdots T} \mathbb{P}(U_j \mid \mathcal{M}) \log(\frac{\mathbb{P}(U_j \mid \mathcal{M})}{\mathbb{Q}(U_j)}) + (1-s)\log(\frac{1-s}{1-p}) \\
&= \mathbb{E}_{\mathbb{P}(\mathcal{M})}(1-s)\log \frac{1-s}{1-p} + s\sum_{i=1}^{T} q_i \log \frac{sq_i}{pm_i}
\end{aligned}
\tag{12}
$$

Combining Eq. 10 and Eq. 12, we obtain the following optimization objective:

$$
\min_{\phi} \mathbb{E}_{e \in \mathcal{E}(t)} \sum_{c=0,1} -\mathbb{1}(Y_f[e] = c)\log(f(\mathcal{G}_{\text{exp}}^e)[e]) + \beta((1-s)\log \frac{1-s}{1-p} + s\sum_{i=1}^{T} q_i \log \frac{sq_i}{pm_i}),
\tag{13}
$$

where $\phi$ denotes learnable parameters in TempME, $\beta$ is a regularization coefficient.

## D   Proposed Approach

### D.1   Null Model

The analysis of temporal motif distribution is typically presented in terms of a null model [53, 22]. A null model is essentially a randomized version of the empirical network, generated by shuffling or randomizing certain properties while preserving some structural aspects of the original graph. The null model serves as a baseline against which the observed motif distribution can be compared, allowing us to evaluate the presence of meaningful patterns and deviations from randomness. By comparing the motif distribution of the empirical network with that of the null model, we can distinguish between motifs that arise due to non-random structural or temporal features of the network from those that are simply a result of random processes. Following prior works on the null model [67, 22], we utilize the most obvious null model in this work, where the event order is randomly shuffled. Formally, a temporal graph $\mathcal{G} = \{\mathcal{V}(t), \mathcal{E}(t)\}$ can be defined as a sequence of interaction events, *i.e.*, $\mathcal{G} = \{(u_i, v_i, t_i)\}_{i=1}^{N}$, where $u_i$ and $v_i$ are two nodes interacting at time $t_i$. We generate a permutation $\sigma \in S_N$, where $N$ refers to the number of interaction events within the temporal graph $\mathcal{G}$. Random-ordered graph $\mathcal{G}_\sigma$ is then constructed by $\mathcal{G}_\sigma = \{u_i, v_i, t_{\sigma(i)}\}_{i=1}^{N}$. In this way, the topological structure and degree spectrum of the temporal network are kept, while the temporal correlations are lost.

### D.2   Equivalence Class

Two temporal motifs with the same digital representations are equivalent according to Definition B. With the $2l$-digit representations, we can easily classify all temporal motif instances according to

their equivalence class relations. For example, with up to 3 nodes and 3 events, there are twelve equivalence classes as shown in Figure 5 (a). In essence, the digital representation provides an efficient way to grow a motif. For instance, in Figure 5 (a), the last three columns of each row can be considered as the results of generating a new interaction event based on the motif in the first column. When it comes to motifs with 4 events, we attach new digit pairs to motifs with 3 events. For example, the next digit pair is one in $\{01, 02, 03, 12, 13, 23\}$ for the temporal motif assigned with $010201$.

Marginal distributions of temporal motifs reveal the governing patterns and provide insights into a better understanding of the dynamics in temporal graphs. To obtain comprehensive knowledge about the distribution of these temporal motifs, we sample a fixed number of temporal motifs around each node in a temporal graph and the corresponding null model. We compute the average probability of each equivalence class across all nodes and visualize the results in Figure 5 (b). We can observe that the difference between the empirical distribution of the temporal motifs and that of its randomized version varies significantly across different datasets. For example, the proportion of motif $010101$, which corresponds to repeated interactions between two nodes, deviates significantly from randomness in Wikipedia, indicating its importance to the underlying dynamics. Moreover, results on the null model reveal that the motif $0112$ is mostly a natural result of random processes.

### D.3 Efficient Sampling and Complexity Anlaysis

The temporal motif sampling algorithm is given in Alg. 2. The brute-force implementation of Alg. 2 results in the time complexity of $\mathcal{O}(Cl)$, which can be further decreased with tree-structured sampling. We discuss two cases. Firstly, when $n \geq l+1$, there is actually no constraint on the number of nodes within each temporal motif. We create a sampling configuration $[k_1, k_2, \cdots, k_l]$ satisfying $\sum_{i=1}^{l} k_i = C$. It indicates that we sample $k_1$ events starting from $u_0$ at the first step and then sample $k_2$ neighboring events for each of the $k_1$ motifs sampled in the previous step. Repeat the step for $l$ times and we obtain $\sum_{i=1}^{l} k_i = C$ temporal motifs in total. Secondly, if $n \leq l$ (*i.e.,* Alg. 1), we create a sampling configuration

---

**Algorithm 2:** Temporal Motif Sampling $(\mathcal{E}, n, l, u_0, t_0, C)$

---

**1** Node set: $S_c \leftarrow \{u_0\}$, for $1 \leq c \leq C$
**2** Event sequence: $I_c \leftarrow ()$, for $1 \leq c \leq C$
  **for** $c = 1$ *to* $C$ **do**
    **for** $j = 1$ *to* $l$ **do**
**3**      Sample one event $e_j = (u_j, v_j, t_j)$
        from $\mathcal{E}(S_c, t_{j-1})$
        **if** $|S_c| < n$ **then**
**4**        $S_c = S_c \cup \{u_j, v_j\}$
          $I_c = I_c \parallel e_j$

**return** $\{I_c \mid 1 \leq c \leq C\}$

---

$[k_1, k_2, \cdots, k_{n-1}]$ satisfying $\sum_{i=1}^{n-1} k_i = C$. Similarly, we sample $k_i$ neighboring events at the $i$-th step. We repeat the process for $n-1$ times and obtain $C$ temporal motifs with $n-1$ events in total. For each of the $C$ temporal motifs, we sample a neighboring event for $l-n+1$ times and ensure the number of nodes in each temporal motif is no more than $n$, which completes the $n-1$-length temporal motif to involve $l$ events in total. The upper bound time complexity of the tree-structured sampling is $\mathcal{O}(C(l-n+2))$. Specifically, when $n \geq l+1$, the time complexity is reduced to $\mathcal{O}(C)$. We use the $2l$-digit to represent the temporal motifs. Two temporal motifs that have the same $2l$-digit are equivalent. The equivalence classification results in the time complexity of $\mathcal{O}(C)$. For event anonymization, we first identify unique node pairs $(u_i, v_i)$ that occur in $C$ temporal motifs and count their occurrence times at each position $j$, where $j = 1, \cdots, l$. Then we utilize the position-aware counts to create the structural features for each event in the temporal motifs. This process results in the time complexity of $\mathcal{O}(Cl)$.

On the other hand, the previous TGNNExplainer requires re-searching individually for each given instance. To infer an explanation for a given instance, the time complexity of TGNNExplainer with navigator acceleration is $\mathcal{O}(NDC)$, where $N$ is the number of rollouts, $D$ is the expansion depth of each rollout and $C$ is a constant including inference time of navigator and other operations.

### D.4 Model Details

By default, we use GINE [74] as the MESSAGEPASSING function and Mean-pooling as the READOUT function. GINE convolution adapts GIN convolution to involve edge features as follows,

$$x_i' = h_{\theta_1}((1 + \epsilon) \cdot x_i + \sum_{j \in \mathcal{N}(i)} \text{ReLU}(x_j + h_{\theta_2}(E_{ji}))), \quad (14)$$

where $h_{\theta_1}$ and $h_{\theta_2}$ are neural networks. $x_i$ and $E_{ji}$ denote the features of node $i$ and edge $j \sim i$, respectively. $E_{ji} = (a_{ji} \parallel T(t - t_{ji}) \parallel h(e_{ji}))$ contains the associated event attributes, time encoding and structural features. $\mathcal{N}(i)$ refers to the neighboring edges of node $i$. Mean-pooling takes the average of the features of all nodes within the graph and outputs a motif-level embedding.

# E   Experiments

## E.1   Dataset

We select six real-world temporal graph datasets to validate the effectiveness of TempME. These six datasets cover a wide range of real-world applications and domains, including social networks, political networks, communication networks, *etc.*The brief introduction of the six datasets is listed as follows. Data statistics are given in Table 7.

- Wikipedia [75] includes edits made to Wikipedia pages within a one-month period. The nodes represent editors and wiki pages, while the edges represent timestamped posting requests. The edge features consist of LIWC-feature vectors [76] derived from the edit texts, each with a length of 172.

- Reddit dataset [75] captures activity within subreddits over a span of one month. In this dataset, the nodes represent users or posts, and the edges correspond to timestamped posting requests. The edge features consist of LIWC-feature vectors [76] extracted from the edit texts, with each vector having a length of 172.

- Enron [77] is an email correspondence network where the interaction events are emails exchanged among employees of the ENRON energy company over a three-year period. This dataset has no attributes.

- UCI [78] is an unattributed social network among students of UCI. The datasets record online posts on the University forum with timestamps with the temporal granularity of seconds.

- Can.Parl. [79] is a dynamic political network that tracks the interactions between Canadian Members of Parliament (MPs) spanning the years 2006 to 2019. Each node in the network represents an MP who represents an electoral district, while the edges are formed when two MPs both vote "yes" on a bill. The weight assigned to each edge reflects the number of times an MP has voted "yes" for another MP within a given year.

- US Legis [79] focuses on the social interactions among legislators in the US Senate through a co-sponsorship graph. The edges in the graph correspond to the frequency with which two congresspersons have jointly sponsored a bill during a specific congressional session. The weight assigned to each edge indicates the number of times such co-sponsorship has occurred. The dataset records the interactions within 12 congreessions.

Table 7: The dataset statistics. Average interaction intensity is defined as $\lambda = 2|E|/(|V|T)$, where $E$ and $V$ denote the set of interactions and nodes, $T$ is the dataset duration in the unit of seconds.

| Datasets | Domains | #Nodes | #Links | #Node&Link Features | Duration | Interaction intensity |
|---|---|---|---|---|---|---|
| Wikipedia | Social | 9,227 | 157,474 | 0&172 | 1 month | $1.27 \times 10^{-5}$ |
| Reddit | Social | 10,984 | 672,447 | 0&172 | 1 month | $4.57 \times 10^{-5}$ |
| Enron | Communication | 184 | 125,235 | 0&0 | 1 month | $1.20 \times 10^{-5}$ |
| UCI | Social | 1,899 | 59,835 | 0&0 | 196 days | $3.76 \times 10^{-6}$ |
| Can.Parl. | Politics | 734 | 74,478 | 0&1 | 14 years | $4.95 \times 10^{-7}$ |
| US Legis | Politics | 225 | 60,396 | 0&1 | 12 congresses | - |

None of the used temporal graphs contains node attributes. Wikipedia and Reddit have rich edge attributes. Can.Parl. and US Legis contain a single edge attribute, while Enron and UCI contain no edge attribute. All datasets are publicly available at `https://github.com/fpour/DGB`.

## E.2   Experimental Setup

**Base Model**. There are two main categories of Temporal GNNs [7]. One type utilizes local message passing to update the time-aware node features (MP-TGNs). The other type aggregates temporal

walks starting from the nodes of interest and leverages RNNs to learn the sequential information (WA-TGNs). Since we aim at identifying the most explanatory historical events and their joint effects in the form of a temporal motif, we focus on MP-TGNs in this work.

We adopt three popular state-of-the-art temporal GNNs that augment local message passing, TGAT [3], TGN [4] and GraphMixer [5]. TGAT[2] aggregates temporal-topological neighborhood features and time-feature interactions with a modified self-attention mechanism. TGN[3] incorporates a memory module to better store long-term dependencies. Both TGAT and TGN leverage a time encoding function to effectively capture the time-evolving information contained in the interactions. GraphMixer[4] is one of the newest frameworks that utilize an MLP-Mixer [80]. On the contrary, GraphMixer uses a fixed time encoding function, showing great success and potential.

**Configuration**. We follow the standard dataset split for temporal graphs [1, 6]. We sort and divide all interaction events by time into three separate sets, corresponding to the training set, validation set and testing set. The split points for the validation set and testing set are $0.75T$ and $0.8T$, where $T$ is the duration of the entire dataset. We keep the inductive setting for all three base models. We mask $10\%$ nodes and the interactions associated with the masked nodes during the training stage. For evaluation, we remove all interactions not associated with the masked nodes, so as to evaluate the inductiveness of temporal GNNs.

**Results**. Hyperparameters are sufficiently fine-tuned for each base model. The number of attention heads is tuned in $\{1, 2, 3\}$ and the number of layers in $\{1, 2, 3\}$ for TGAT and TGN. The number of MLPMixer layers is 2 for GraphMixer. The dimension of the time encoding is 100, and the output dimension is 172. The maximum number of epochs is 50. An early stopping strategy is used to mitigate overfitting. The link prediction results (Average Precision) in the inductive setting are shown in Table 8.

Table 8: Link prediction results (Average Precision) of TGAT, TGN and GraphMixer

|  | Wikipedia | Reddit | UCI | Enron | USLegis | Can.Parl. |
|---|---|---|---|---|---|---|
| **TGAT** | 93.53 | 96.87 | 76.28 | 65.68 | 72.35 | 65.18 |
| **TGN** | **97.68** | **97.52** | 75.82 | **76.40** | **77.28** | 64.23 |
| **GraphMixer** | 96.33 | 95.38 | **89.13** | 69.42 | 66.71 | **76.98** |

There exist some differences between Table 8 and the results in the original paper of GraphMixer. We ascribe the difference to our inductive setting. In the original paper of GraphMixer, they conduct experiments under the transductive learning setting. To keep consistent, we follow [73] to remove the one-hot encoding of the node identities in GraphMixer to ensure the inductivenss, which potentially results in performance degradation in some cases.

**Baselines**. We consider the following baselines:

- **ATTN** leverages the internal attention mechanism to characterize the importance of each interaction event. The intuition is that events that are assigned with larger attention values are more influential to the model's prediction. We adopt ATTN to explain the predictions made by TGAT and TGN, since they both involve attention layers as their building blocks.

- **Grad-CAM** was originally proposed in computer vision to identify the most significant patches in an image. [37] proposes to generalize Grad-CAM to the discrete graph domain. We compute the gradient of the model's output *w.r.t.* event features and take the norm as the importance score for the corresponding event.

- **GNNExplainer** [13] is a learning-based method that optimizes continuous masks for neighboring events upon which a base model makes the prediction. We follow the same setting as proposed in the original paper.

- **PGExplainer** [14] trains a deep neural network that generates continuous masks for the input graph. Event feature is defined as $E_i = (a_i \| T(t - t_i) \| h(e_i))$, the same as the one used in our Temporal Motif Encoder (Eq. 3). PGExplainer takes as input the event features and outputs a continuous mask for each neighboring event.

---

[2]https://github.com/StatsDLMathsRecomSys/Inductive-representation-learning-on-temporal-graphs
[3]https://github.com/yule-BUAA/DyGLib
[4]https://github.com/yule-BUAA/DyGLib

- **TGNNExplainer** [23] is a searching-based approach that leverages the Monte Carlo Tree Search algorithm to explore effective combinations of explanatory events. We follow the same training procedure as proposed in the original paper.

**Implementation Details**. The implementation and training are based on the NVIDIA Tesla V100 32GB GPU with 5,120 CUDA cores on an HPC cluster. The learning rate is initially set as $1e-3$ and batch size is set as $64$. The maximum training epochs is 100. We summarize the search ranges of other main hyperparameters used in TempME in Table 9.

Table 9: Search ranges of hyperparameters used in TempME

|  | # temporal motifs per node $C$ | Beta $\beta$ | Prior Belief $p$ |
|---|---|---|---|
| Wikipedia | $\{20, 30, 40, 50, 60\}$ | $\{0.2, 0.4, 0.6, 0.8, 1\}$ | $\{0.1, 0.2, 0.3, 0.4, 0.5\}$ |
| Reddit | $\{20, 40, 60, 80, 100\}$ | $\{0.2, 0.4, 0.6, 0.8, 1\}$ | $\{0.1, 0.2, 0.3, 0.4, 0.5\}$ |
| UCI | $\{20, 30, 40, 50, 60\}$ | $\{0.2, 0.4, 0.6, 0.8, 1\}$ | $\{0.2, 0.3, 0.4, 0.5, 0.6, 0.8\}$ |
| Enron | $\{20, 30, 40, 50, 60\}$ | $\{0.2, 0.4, 0.6, 0.8, 1\}$ | $\{0.1, 0.2, 0.3, 0.4, 0.5\}$ |
| USLegis | $\{20, 30, 40, 50, 60\}$ | $\{0.2, 0.4, 0.6, 0.8, 1\}$ | $\{0.2, 0.3, 0.4, 0.5, 0.6, 0.8\}$ |
| Can.Parl. | $\{20, 30, 40, 50, 60\}$ | $\{0.2, 0.4, 0.6, 0.8, 1\}$ | $\{0.2, 0.3, 0.4, 0.5, 0.6, 0.8\}$ |

### E.3  Motif-enhanced Link Prediction

Table 10 shows the complete results on link prediction enhancement with motif embedding. The performance boosts on Wikipedia and Reddit are relatively limited, due to the exceedingly high performance achieved by base models. However, motif embedding demonstrates the ability to greatly improve the link prediction performance on more challenging datasets, *e.g.,* USLegis, Can.Parl. We

Table 10: Link prediction results (Average Precision) of base models with Motif Embedding (ME)

|  | Wikipedia | Reddit | UCI | Enron | USLegis | Can.Parl. |
|---|---|---|---|---|---|---|
| TGAT | 93.53 | 96.87 | 76.28 | 65.68 | 72.35 | 65.18 |
| TGAT+ME | $\mathbf{95.12}^{(\uparrow 1.59)}$ | $\mathbf{97.22}^{(\uparrow 0.35)}$ | $\mathbf{83.65}^{(\uparrow 7.37)}$ | $\mathbf{68.37}^{(\uparrow 2.69)}$ | $\mathbf{95.31}^{(\uparrow 22.96)}$ | $\mathbf{76.35}^{(\uparrow 11.17)}$ |
| TGN | 97.68 | 97.52 | 75.82 | $\mathbf{76.40}$ | 77.28 | 64.23 |
| TGN+ME | $97.68^{(\uparrow 0.00)}$ | $\mathbf{98.35}^{(\uparrow 0.83)}$ | $\mathbf{77.46}^{(\uparrow 1.64)}$ | $75.62^{(\downarrow 0.78)}$ | $\mathbf{83.90}^{(\uparrow 6.62)}$ | $\mathbf{79.46}^{(\uparrow 15.23)}$ |
| GraphMixer | 96.33 | 95.38 | 89.13 | 69.42 | 66.71 | 76.98 |
| GraphMixer+ME | $\mathbf{96.51}^{(\uparrow 0.18)}$ | $\mathbf{97.81}^{(\uparrow 2.43)}$ | $\mathbf{90.11}^{(\uparrow 0.98)}$ | $\mathbf{70.13}^{(\uparrow 0.71)}$ | $\mathbf{81.42}^{(\uparrow 14.71)}$ | $\mathbf{79.33}^{(\uparrow 2.35)}$ |

can also notice that the performance improvement on Enron is not as significant as other datasets. The reason is that there are multiple identical interactions in the Enron dataset. On average, each distinct interaction is accompanied by $3.284$ exactly identical interactions within this dataset, far more than other datasets. While this phenomenon might be deemed reasonable within the context of email networks (*e.g.,* the Enron dataset), wherein multiple emails are dispatched to the same recipient at identical timestamps, it is not a common phenomenon in other datasets and many real-world scenarios. For consistency with existing literature [22, 24] we restrict the timestamp of the next sampled event strictly earlier than the previous event, which is also a necessary condition of underlying causality between interactions. Consequently, many identical interactions in the Enron dataset are not sampled within a temporal motif, thereby potentially degrading the performance improvement of TempME over this specific dataset. As a result, one limitation of TempME is analyzing temporal graphs characterized by high interaction density between the same node pairs at the same timestamp.

### E.4  Runtime Evaluation

To empirically verify the time complexity and efficiency of the proposed TempME, we test sampling and encoding runtime *w.r.t.* number of temporal motifs and length of temporal motifs, as shown in Figure 6. The base model is set as TGAT and the dataset is Reddit, which is a massive and large-scale dataset. *Encoding* process includes the temporal motif encoding and the following MLP to generate the importance scores. The averages and standard deviations are calculated across all target events. The maximum number of nodes within each temporal motif is set to the length of the temporal motifs, *i.e.,* $n = l$. We can make the following observations from Figure 6. (1) The runtime of sampling is

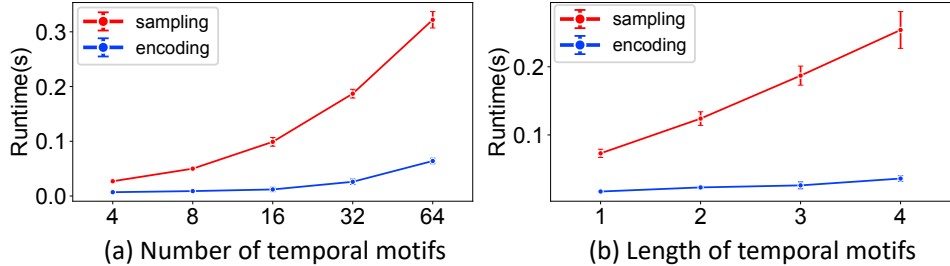

(a) Number of temporal motifs    (b) Length of temporal motifs

Figure 6: (a) Sampling and encoding runtime *w.r.t.* the number of temporal motifs around each node (b) Sampling and encoding runtime *w.r.t.* length of temporal motifs

much longer than that of the encoding process under the same number of motifs and motif length, especially with a larger number of temporal motifs. (2) The runtime of sampling and encoding is approximately in proportion to the length of temporal motifs. The runtime comparison between TempME and baselines is shown in Table 3.

## E.5 Insights from Explanations

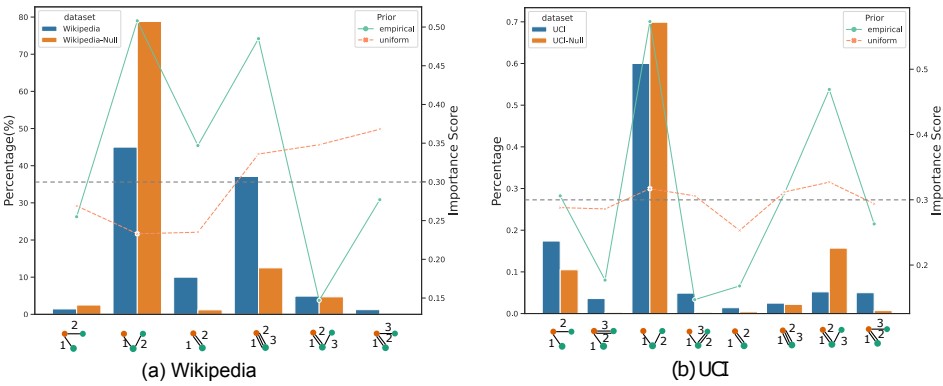

(a) Wikipedia    (b) UCI

Figure 7: Occurrence percentages of temporal motifs in empirical graph and its null model (only a subset of the most frequent motifs are displayed). The lines represent the average importance score across all instances of motifs within the respective motif class.

The use of an information bottleneck framework in a simplistic manner can introduce a bias that favors specific motifs, potentially leading to the oversight of less frequent yet important temporal patterns. Introducing the null model, on the other hand, has a primary impact of shifting the focus from absolute proportions to relative proportions compared to the null model itself. This shift helps alleviate attention toward motifs that offer limited information when evaluated solely based on their frequency. Figure 7 illustrates the visualization of importance scores for temporal motifs and their corresponding occurrence percentages. In both prior distributions, the prior belief $p$ is set to $0.3$. The dashed gray line represents an importance score of $0.3$. When employing the *uniform* prior distribution, the model tends to assign all interaction-irrelevant motifs the prior belief $p$, regardless of their varying occurrence percentages. Conversely, the *empirical* prior distribution takes the null model into consideration and highlights motifs that convey more information based on their occurrence percentages. Consequently, the *empirical* prior distribution leads to a higher deviation from the gray horizontal line, while the average importance score remains close to the prior belief $p$. By considering the null model, a more comprehensive analysis becomes possible, encompassing the significance and uniqueness of the observed motifs beyond their raw occurrence probability. For instance, in the Wikipedia dataset, the *empirical* prior distribution captures the second and fourth motifs with high importance scores, as displayed in Figure 7(a). Both motifs exhibit a significant difference in their occurrence probability compared to that in the null model. Conversely, in the UCI dataset, the *uniform* prior distribution yields a relatively uniform distribution of importance scores across motifs, thereby providing limited information regarding the distinct contributions of these temporal motifs.

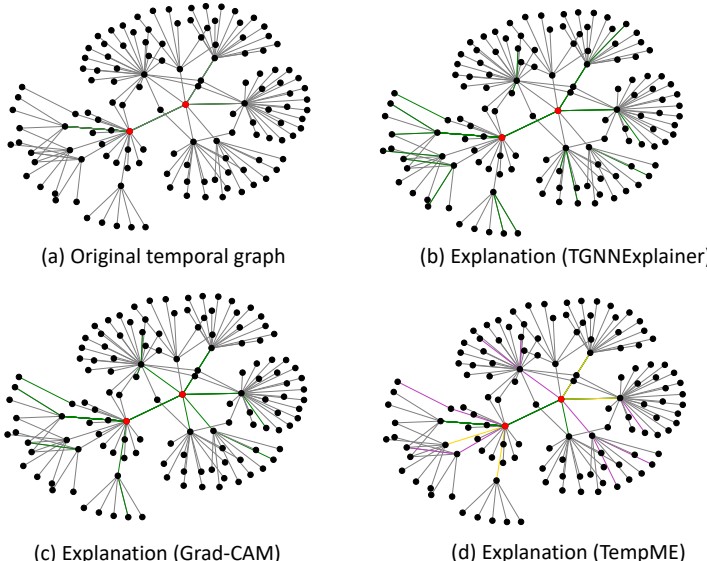

(a) Original temporal graph        (b) Explanation (TGNNExplainer)

(c) Explanation (Grad-CAM)        (d) Explanation (TempME)

Figure 8: (a) Original temporal graph. (b) Explanation example generated by TGNNExplainer. (c) Explanation example generated by Grad-CAM. (d) Explanation example generated by TempME. The link between the two red nodes is to be explained. The explanations (*i.e.,* explanatory edges) are highlighted in colors.

Figure 8 shows the explanation examples generated by TGNNExplainer, Grad-CAM and TempME on Wikipedia. The base model is TGAT with two layers. Red nodes indicate two ends of the event to be explained. All graphs are constrained within the 2-hop neighbor of the two red nodes. In Figure 8(d), different colors represent the different types of temporal motifs the corresponding event contributes to. Compared with TGNNExplainer, TempME works better in generating a *cohesive* explanation. Moreover, the explanation generated by TempME provides additional motif-level insights, thus being more human-intelligible.

# F Discussion

**Limitation**. While employing temporal motifs to generate explanations shows promise, several limitations need to be acknowledged. First, the identification of influential motifs relies on the assumption that motifs alone capture the essential temporal dynamics of the graph. However, in complex real-world scenarios, additional factors such as external events, context, and user preferences may also contribute significantly to the explanations. Second, the scalability of motif discovery algorithms can pose challenges when dealing with large-scale temporal graphs. Finally, the selection of a null model may also introduce inductive bias to the desired explanations. Further analysis of the null model setting will be one of the future directions.

**Broader Impacts**. By enabling the generation of explainable predictions and insights, temporal GNNs can enhance decision-making processes in critical domains such as healthcare, finance, and social networks. Improved interpretability can foster trust and accountability, making temporal GNNs more accessible to end-users and policymakers. However, it is crucial to ensure that the explanations provided by the models are fair, unbiased, and transparent. Moreover, ethical considerations, such as privacy preservation, should be addressed to protect individuals' sensitive information during the analysis of temporal graphs.

**Future Works**. In the future, several promising directions can advance the use of temporal motifs proposed in this work. First, incorporating external context and domain-specific knowledge can enhance the explanatory power of motif-based explanations. This can involve integrating external data sources, leveraging domain expertise, or considering multi-modal information. Moreover, developing scalable motif discovery algorithms capable of handling massive temporal graphs will facilitate the applicability of motif-based explanations in real-world scenarios.

