# OpenReview forum: "TempME: Towards the Explainability of Temporal Graph Neural Networks via Motif Discovery"
_NeurIPS.cc/2023/Conference — NeurIPS 2023 poster_

### Official Review · Reviewer_U22B · 2023-06-24

**Soundness:** 3 good
**Presentation:** 3 good
**Contribution:** 3 good
**Rating:** 6
**Confidence:** 4

**Summary:**

The work presents TempME, an explanation methodology for temporal graph neural networks that incoporates the idea of temporal motifs. By incoporating temporal motifs, the method demonstrates better cohesiveness, since motifs are constructed to be cohesive, i.e., connected and localized, as well as better explainability compared to prior explainability works.


**Strengths:**

- The work presents an excellent contribution to TGNN explainability by proposing the use of temporal motifs.
- The work presents solid technical material to support the proposed methodology.
- The work further presents good and extensive results that covers many interesting aspects.



**Weaknesses:**

There are some unclarity in the paper. In particular, the goal of the evaluation and the evaluation setting for the explanation performance is not clear. In particular, the statement "ACC-AUC, which is the AUC value of the proportion of generated explanations that have the same predicted label by the base model over sparsity levels from 0 to 0.3" is hard to comprehend. Currently, the reviewer thinks that:
- "generated explanations" are the explanation subgraphs extracted by the expalantion models.
- "sparcity level" idicates the ratio of sizes between the the extracted explanation subgraph and the whole temporal graph
- perhaps the "predicted label by the base model" is the prediction of the event/edge that the original target model gives when given the extracted explanation subgraph?

It could be clearer if at Section 3, it is noted that the budget "K" will be represented as "sparsity" and the objective of Equation 1 is spelled out more intuitively.

Besides:
- The inline equation for "Fidelity" has typos and the paranthesis does not match up. It also lacks intuitive explanations. Besides, it seems that Fidelity follows that constructed by TGNNExplainer, and a proper citation should be given.
- norm of the gradient being utilized for Grad-CAM is essential to understanding the treatment of baseline and should be stated in the main paper.

Miscellaneous:
The figures (especially the color shades and the lighter colors, when printed out) is not clear.


**Questions:**

Regarding the motivation for presenting an explanation subgraph for TGNN: While with image instances we can verify that the important regions (e.g. highlighted by grad-cam) coincide with semantically meaningful regions in an image using human intuition and understanding, it is unclear if human intuition or logic can be applied to temporal graphs.
- Can you provide examples of  human intuition or logic applied to temporal graphs? For example, with certain datasets, observing some part of the temporal graph would lead to a understanding on whether some target event should or should not exist.
- Can you provide examples of the explanation subgraph presented by TempME and show how such explanation subgraph provides "human intelligible explanation" as stated in line 38?
- Can you provide examples of the explanation subgraph presented by TGNNExplainer and Grad-CAM, as well as other baselines, to show how they do not present "human intelligible explanation"?


**Limitations:**

The paper presents some technical limitations but the discussions on potential negative societal impact could be strengthened by discussing the potential real-world application of explaining TGNNs.

---

> ### Author Rebuttal · Authors · 2023-08-10
>
> Thanks for your valuable feedback, which significantly improves the quality of this work. We would like to address the following potential concerns you raise.
>
> >The goal and setting of the evaluation are not clear. Clarification to “AUC value of the proportion of generated explanations that have the same predicted label by the base model over sparsity levels from 0 to 0.3”?
>
> Sorry for any confusion. The sparsity level indicates the ratio of sizes between the extracted explanation and the computational graph of the event to explain. All of your other understanding is correct. To explain more, the accuracy of explanations is sensitive to the sparsity level. The area under the accuracy-sparsity curve is denoted as the ACC-AUC value, which is reported in Table 1.
>
> >It could be clearer if the budget “K” is represented as “sparsity” in Eq.1 and any intuitive explanations to Eq.1?
>
> Thanks for the constructive comments. We will rewrite the budget $K$ as $|\mathcal{G}_{\exp }^e| \leq s|\mathcal{G}(e)|$, where $s$ is the sparsity level, $\mathcal{G}(e)$ is the computational graph of the event $e$.  Eq(1) aims to maximize the mutual information between the explanation and the original model prediction.
>
> >Lack of intuitive explanations “Fidelity”.
>
> “Fidelity” measures how valid and faithful the explanations are to the model’s original prediction. If the original prediction is positive, then an explanation leading to an increase in the model’s prediction logit is considered to be more faithful and valid. If the original prediction is negative, then an explanation that decreases the prediction logit is better. We will add more intuitive explanations to this metric in our revised version.
>
> >Motivation for presenting an explanation subgraph for TGNN. How can human intuition or logic be applied to temporal graphs? How does TempME provide “human-intelligible explanation” for TGNN? How do other baselines fail to provide “human-intelligible explanations” for TGNN?
>
> Thanks for the constructive feedback. Please refer to the real-world examples we provide in the general response. Explanation visualizations are provided in the 1-page pdf.   Compared with TGNNExplainer, TempME works better in generating a ”cohesive” explanation. Moreover, the explanation generated by TempME provides additional motif-level
> insights. In our visualization, different colors indicate the different types of temporal motifs that the corresponding event contribute to. Temporal motifs work as the building block of the dynamic system and provide a unique perspective of how the combination of events (e.g., preferential attachment, triadic closure, etc) contributes to the prediction of temporal GNNs. In contrast, we observe that previous methods (e.g., GradCAM, TGNNExplainer) either fail to generate a "cohesive" explanation or fail to provide any model-level insight.
>
> > Others: Typos in the inline equation for “Fidelity” / Citation for Fidelity metric / Gradient Norm utilized for Grad-CAM / unclear color in figures
>
> Thanks for your careful reviews. We will add the Grad-CAM details to the main text, cite for fidelity metric, change the figure colors for better readability, and fix the typos accordingly in our revised version.

---

> > ### Comment · Reviewer_U22B · 2023-08-18
> >
> > Thank you. I am satisfied with your explanations. However, I maintain my score at weak accept based on the impact of this paper.

---

### Official Review · Reviewer_NnW1 · 2023-06-26

**Soundness:** 3 good
**Presentation:** 3 good
**Contribution:** 3 good
**Rating:** 7
**Confidence:** 3

**Summary:**

This paper provides a method to extract temporal motifs in order to capture correlated building blocks of networks.

**Strengths:**

The method of extracting temporal motifs is interesting; it is based on information theoretic concepts.

**Weaknesses:**

In my view there are two main weaknesses of this paper.
The first weakness is the underlying assumption that network motifs should consist of nodes and edges which are in close time proximity to each other. This assumption is not questioned but often does not hold, as there are often seasonal effects in network time series. These seasonal effects could occur for example only once per week, as Sundays may be special in terms of network behaviour.

The second weakness is the presentation. It is stated that the method is in a continuous-time setting, but the events occur at separated time points from the description; there does not seem to be the possibility of having more than one event occurring simultaneously. The large number of questions below indicate that the paper may not be very clearly written.

The paper aims to provide an explanation of temporal GNNs; it would be good to see a worked example in which explanation relates to a tangible real-world scenario. Which are the motifs that are found in the different networks? How can they be viewed as builing blocks of these networks; do they point towards fundamental differences between, say, the Wikipedia network and the Enron network?  There is a start at this question in the supplementary material E.5 but the analysis may strongly depend on the chosen prior.

Finally, the discussion from the supplementary information Section F should have been in the main text.

**Questions:**

It is not clear to me what the mutual information is between a random variable and a graph.

Are temporal motifs allowed to overlap each other?

Why is there no time component in Figure 1?

The elements in the edge set contain an event attribution but the motif definition ignores this event attribution. Why?

In Definition 1 why is u_0 = u_1 set ? What is the induced subgraph? What is the graph that this is a subgraph of? If this underlying graph is the union of all temporal graphs, then does it have multiple edges or self-loops?

Definition 2: When do two motif instances have the same topology? Is it their induced subgraph that has the same topology? Again, what happens to mutiple edges?

Why can the motif of preferential attachment not be represented as a temporal walk?

What are `surrounding' motifs in Section 4.2.?

Temporal motif encoding: what is T ( t - t_{pq})  ? What is N(p)? Is  there an underlying graph with a fixed number of nodes and a fixed topology? Is that a simple graph?

How is an importance score generally obtained? It is a crucial ingredient of the method.

In Section 5.2 it is stated that ``TempME still surpasses all baselines and achieves the highest connectivity levels, primarly due to it ability to extract and utilize self-connected motifs'' ; how does self-connectedness enter here? what it meant by that?

Why is the discussion from the supplementary information section F not in the main text? Is it not important?



**Limitations:**

The paper discusses limitations  only in the supplementary material. Clearly the choice of null model may have a major influence on the analysis, as does the choice of prior.

---

> ### Author Rebuttal · Authors · 2023-08-10
>
> Thanks for your valuable feedback and positive comments on the theoretical contribution. We address below the questions in order.
>
> > The assumption that ...  may not hold when there are seasonal effects.
>
> We also believe that considering these seasonal effects is crucial. However, we think developing seasonal motifs might deserve another paper. First, the definition of temporal motifs that represent seasonal effects is nontrivial, since we might need to first recognize the most significant time intervals that reflect the system dynamics. Second, given a time interval, how to sample the target seasonal motifs efficiently is also under-explored, which requires extensive effort. We sincerely appreciate your valuable comments, which actually light up an interesting future direction for us.
>
> > The method is in a continuous-time setting, but the events occur at separated time points. There does not seem to be the possibility of having more than one event occurring simultaneously.
>
> Sorry for any confusion. Actually, the continuous-time setting we used in this manuscript supports simultaneous events. “Continuous-time” means the timestamp($t_k$) in the event element can be any continuous value in the real number set $\mathbb{R}$, rather than being restricted to a specific value derived from a sequence of time points with a constant time interval. In a continuous-time setting, there can be multiple simultaneous events with the same value of timestamp.
>
> > It would be good to see a worked example related to real-world scenarios.
>
> Thanks for the constructive feedback. Please refer to the real-world example we provide in the general response.
>
> >What is the mutual information between a random variable and a graph?
>
> A graph is also a variable represented by its adjacency matrix, and node/edge attributes. Since the graph follows a certain distribution (either a prior distribution given the dataset, or a posterior distribution predicted by networks), we can calculate the mutual information between a random variable and this graph random variable.
>
> >Are temporal motifs allowed to overlap each other?
>
> Yes. They can overlap each other.
>
> >Why no time component in Figure 1?
>
> Time components are included in Figure 1. Here “1,2,3…” indicates the order of event occurrence.
>
> >Why does motif definition ignore the event attribution?
>
> Network motifs primarily focus on the arrangement and connectivity patterns of nodes within a network. In this work, we also follow this idea for consistency with existing literature.
>
> >Why $u_0=u_1$ in Def. 1? What does “induced subgraph” mean? any multiple edges or self loops?
>
> It means we sample motifs starting from the given node of interest. Therefore, the starting node of the first event should be the same as the given node. “Induced subgraph” means the graph formed from all vertices and events in $I$, which is the subgraph of the original temporal graph. There can be multiple edges between two nodes. Typically, in network motif studies, input networks are considered after all self-loops are discarded.
>
> >What does "the same topology” mean? What about multiple edges?
>
> “The same topology” means the induced subgraph has the same topology and the corresponding events happen in the same order over time (regardless of the absolute time interval). We use a $2l$-digit to represent the identities of sampled nodes in the event order. “Two motif instances are the same” indicates the same $2l$-digit representation. It is able to manage multiple edges. We discussed more details in Appendix B and D.2.
>
> >Why cannot the preferential attachment ... as a temporal walk?
>
> In Temporal Walk Extraction (Alg.1 in [1]), the next event is sampled from historical events that interact with the latest sampled node. For example, we have sampled $w_1\rightarrow w_2\rightarrow w_3$ that satisfies $w_1\neq w_3$ in the current node set. To extract a motif for preferential attachment, the next event $(u, v, t)$ should satisfy $u=w_2$. However, since the latest sampled node is $w_3$, then $(u,v,t)$ is not in a valid historical event that interacts with the latest node. Therefore, the motif of preferential attachment cannot be sampled by the temporal walk extraction algorithm.
>
> [1] Inductive Representation Learning in Temporal Networks via Causal Anonymous Walks
>
> >What are “surrounding” motifs in Section 4.2?
>
> We sample $C$ motifs starting from $u$ and from $v$ by Algorithm 1 respectively, which represent the surrounding topology.
>
> >What is $T(t-t_{pq})$? What is $N(p)$?
>
> $t$ is the timestamp of the event to explain, and $t_{pq}$ is the timestamp of the historical event $e_{pq}$ in the temporal motif. $T(\cdot)$ is a time encoder that maps the time interval into a 2d-dimensional vector (Line215). $N(p)$ indicates the set of neighboring nodes of $p$. Here, it is essentially the set of nodes in the motif that have interaction with node $p$.
>
> >Is there an underlying graph with a fixed number of nodes and a fixed topology? Is that a simple graph?
>
> No. We are not assuming an underlying graph with a fixed number of nodes/topology/a simple graph.
>
> >How is an important score generally obtained?
>
> Given the motif embedding $m_I$ learned by Temporal Motif Encoder (Eq.3), we adopt an MLP for mapping $m_I$ to an importance score $p_I\in[0,1]$ as mentioned in Line 230-231, which is later used to model the distribution of the corresponding motif.
>
> >What does “self-connectedness” mean?
>
> “Self-connectedness” means each motif sampled by TempME is self-connected. It is guaranteed by the sampling algorithm. The next event during motif sampling is from a set of historical events that interact with at least one node in the current node set. Thereby, the self-connectedness of each motif potentially ensures a high “cohesive” level in the generated explanation.
>
> >Discussion from the supplementary material should be in the main text.
>
> We will adopt your suggestions to move part of the discussions to the main text in our revised version.

---

> > ### Comment · Reviewer_NnW1 · 2023-08-11
> >
> > Thank you. I am satisfied with your explanations.

---

### Official Review · Reviewer_HiZ7 · 2023-06-27

**Soundness:** 3 good
**Presentation:** 4 excellent
**Contribution:** 3 good
**Rating:** 6
**Confidence:** 4

**Summary:**

TempME is an inductive explainer for temporal graph neural networks (TGNNs) over link prediction tasks. It explains TGNNs using temporal motifs, guaranteeing temporally proximate and spatially adjacent explanations, hence more human interpretable. TempME’s pipeline can be broken down into three parts: sampling, embedding, and explaining. Given a link prediction instance, candidate motifs are sampled starting at either end of the edge. The motifs go through a series of steps to generate rich motif embeddings that encode the event's spatial and temporal roles. An MLP learns a Bernoulli distribution over the embeddings. The distribution is used to mask out the best explanation motifs from the candidates. The MLP is trained using the information bottleneck principle to keep the explanations succinct. The mutual information between the predicted label and the explanation is maximized, while the mutual information between the explanation motifs and the set of candidate motifs is minimized. TempME can also boost a model’s performance. The authors show improved results across datasets by concatenating the aggregation of the motif embeddings around a node of interest prior to a model’s final MLP layer.


**Strengths:**

1. Unlike prior work that measures the impact of singular events, TempME measures the combined effect of events on the black box prediction.
2. Cohesive explanations are more human-interpretable than non-cohesive ones.
3. It has significantly less computational cost compared to previous TGNN explainers.
4. It can generalize to unseen nodes, which is highly desirable.
5. The identified motifs can be utilized during training to boost model performance.


**Weaknesses:**

1. The authors have not experimented with synthetic datasets even though they are available. In graphs, the ground truth explanation is often unknown. Hence, synthetic graphs are vital to ascertain that the explanations comply with the ground truth. Please refer to the following for the synthetic datasets and their case study: *Xia, Wenwen, et al. "Explaining temporal graph models through an explorer-navigator framework." The Eleventh International Conference on Learning Representations. 2022.*

2. The authors have cited the following paper as prior work but have not used it as a baseline. *Wenchong He, Minh N Vu, Zhe Jiang, and My T Thai. An explainer for temporal graph neural networks. In GLOBECOM 2022-2022 IEEE Global Communications Conference, pages 462 6384–6389. IEEE, 2022.* Please either compare or provide a justification for not including as a baseline.

3. The authors have compared with GNNExplainer and PGExplainer for comparison with static graph explainers. However, these are no longer state-of-the-art (SOTA) explainers for static graphs. Table 1 and Fig 3 show that GNNExaplainer and PGExplainer are competitive. Using SOTA static graph explainers might have yielded even better results for static explainers and led to different insights. A comparison with the following will be more fruitful:

* *Tan, Juntao, et al. "Learning and evaluating graph neural network explanations based on counterfactual and factual reasoning." Proceedings of the ACM Web Conference 2022. 2022.*

* *Yaochen Xie, Sumeet Katariya, Xianfeng Tang, Edward Huang, Nikhil Rao, Karthik Subbian and Shuiwang Ji. Task-agnostic graph explanations. NeurIPS, 2022.*

4. Please provide the hardware details of the experimental setup.
5. Please provide TempME’s training time as well. It is good to have fast inference, and at the end of the day, inference is what matters, but a user should also have an idea of how long it takes to train.
6. The reference for GINE is correct; however, it does not explicitly use the term GINE in it, which may make it difficult for future readers to refer to. Please add a citation that uses the term GINE explicitly.
7. Pip throws an error when using the author’s requirements.txt to setup the experimental environment.
8. Some typos and grammar mistakes:
 * 158, 622: denotes -> denoted
 * 190: in this work (unnecessary)
 *  278: In meanwhile -> Meanwhile


**Questions:**

Overall, I like the paper. I am open to increasing the score if the authors address the weaknesses listed above and address the questions below.

* Since we are trying to explain the black box and not the data, what if the black box is prioritizing the distant edges? In that case, will TempME force the explanation to be cohesive?
* The authors state the use of a “generative model”. However, it is not found anywhere. The term “generative” brings VAEs and probabilistic graph models to mind, a model that can generate new graph instances that follow a specific distribution or capture the underlying patterns and characteristics of the observed graph data. There is no graph generation involved. Sampling is different from generating. What exactly do the authors mean by a “generative model”? Is it just the ability to generalize to unseen nodes? In that case, the authors should change the terminology as it is misleading.
* In line 194, are C motifs sampled from both u and v, totalling 2C?


**Limitations:**

Yes, the authors have addressed the limitations in the appendix.

---

> ### Author Rebuttal · Authors · 2023-08-10
>
> Thanks for your valuable feedback, which significantly improves the quality of this work. We also address below the potential concerns.
>
> >Lack of Experiments on synthetic datasets
>
> Thanks for your constructive comments. The synthetic dataset is a point process where the arrival of an event can affect the arrival rates of other events. However, there are still no ground-truth explanations from the motif perspective in the synthetic dataset. The relevance and applicability of the point process to real-world temporal graphs have been less tested by the community so far. We believe our extensive experiments on real-world datasets provide a more accurate representation of the challenges that our approach aims to address. We acknowledge the potential value of experiments on synthetic datasets. If time allows, we are more than happy to involve the results on synthetic datasets in our revised version.
>
>
> >Why not include Reference [21] as a baseline?
>
> Previous work [21] is based on a sequence of static snapshots of a temporal graph, which is a very coarse approximation to real-world temporal graphs with continuous time-stamped events and may lose information by looking only at some snapshots of the graph over time. We didn’t involve [21] in our baselines since it might be problematic to compare two models under different problem settings.
>
> >Static graph explainers are no longer state-of-the-art explainers for static graphs. What about more recent explainers?
>
> Thanks for your constructive suggestions. We agree using more SOTA static graph explainers yields stronger experimental results. [1] proposes $CF^2$ that utilizes counterfactual and factual reasoning to generate factual explanations, while guaranteeing the counterfactual property of the remaining subgraphs.
> We refer to the official codes of $CF^2$ and test the explanation performance on Wikipedia and UCI. In our implementation, we generalize the original loss function to explain the link prediction task.  The results are reported as follows. We notice that $CF^2$ archives comparable performance with GNNExplainer, since the intrinsic training procedure of $CF^2$ is the same as GNNExplainer. We consider the additional constraint on the counterfactual property may not hold for temporal link prediction tasks, due to complex time dependencies between events (e.g. one event may have a factual contribution at timestamp $t_1$ but have a counterfactual contribution at timestamp $t_2$)
>
> |            | Wikipedia |        | UCI   |        |
> |------------|------------|--------|-------|--------|
> |            | $CF^2$       | TempME | $CF^2$  | TempME |
> | TGAT       | 84.46      | 85.81  | 73.24 | 76.47  |
> | TGN        | 93.23      | 95.80   | 94.16 | 96.34  |
> | GraphMixer | 89.77      | 90.15  | 60.38 | 87.06  |
>
> Regarding [2], the main focus is explanations for multi-task prediction. The backbone of the proposed explainer shares a very similar spirit with PGExplainer. Thus, [2] cannot be directly compared with our method since we focus on different problems. Meanwhile, we have included PGExplainer as a baseline.
>
> [1]Learning and Evaluating Graph Neural Network Explanations based on Counterfactual and Factual Reasoning
>
> [2]Task-Agnostic Graph Explanations
>
> >Hardware details of the experimental setup
>
> The implementation of our model and training is based on the NVIDIA Tesla V100 32GB GPU with 5,120 CUDA cores on an HPC cluster. The experimental environment is based on  Python 3.8.10, PyTorch 1.10.1 and PyTorch Geometric 2.0.4. We will add hardware details in our revised version.
>
> > Training time of TempME
>
> Thanks for bringing this up. We take Wikipedia as an example (which is a relatively large dataset). We set $C=30, l=3$ in the temporal motif extraction step. It takes around 200 seconds for an epoch and $\approx 60$ epochs to converge. The training time depends on the dataset size. We will discuss this accordingly in our revised version.
>
> >Reference of term GINE
>
> Thanks for the careful suggestions. GINE was first used in [1] as a modified version of GIN to incorporate edge features in the aggregation function. However, we didn’t find any literature that officially proposes the terminology of GINE. We would like to cite both [1] and PyG, which seem to first name this GIN variant as GINE.
>
> [1] Strategies for pre-training graph neural networks
>
> > Experimental environment setup error
>
> Our experimental environment is based on Python 3.8.10. We will refine and update our repository upon publication.
>
> > What if the black box is prioritizing the distant edges? Can TempME still force "cohesive"?
>
> We note that TempME extracts explanatory motifs from the computational graph of a given instance. The number of layers in current temporal GNNs is typically 2 or 3, meaning that the black box considers 2 or 3-hop neighbors. In that case, if the black box is prioritizing the distant edges, TempME is still able to extract a motif that involves the distant edge and the edge to explain, which potentially ensures the “cohesive” property of the explanation.
>
> > Clarification to “generative model”
>
> We agree with your understanding that “a generative model captures a specific distribution and characteristics of the observed graph data and obtains the ability to generate new instances”. TempME is exactly a generative model. We actually frame the explanation task as a generative problem, where the explainer is trained to capture the underlying distribution of explanatory motifs. The importance scores learned by TempME are used to model the Bernoulli distributions of certain motifs. TempME is also able to generate new explanatory subgraphs by sampling from the motif distributions.
>
> > In line 194, are C motifs sampled from both u and v, totalling 2C?
>
> Yes. We sample $C$ motifs for $u$ and $v$ at both ends of the interaction and obtain $2C$ motifs in total.
>
> >Typos/grammar mistakes.
>
> Thanks for pointing them out. We will revise our manuscript accordingly.

---

> > ### Comment · Reviewer_HiZ7 · 2023-08-12
> > **Thank you for the clarifications.**
> >
> > Thank you for the clarifications. I raise my rating to weak accept.

---

### Official Review · Reviewer_M5gq · 2023-07-31

**Soundness:** 3 good
**Presentation:** 3 good
**Contribution:** 2 fair
**Rating:** 5
**Confidence:** 4

**Summary:**

This paper proposes an approach, Temporal Motifs Explainer (TempME), to find out key sub-graph structures from temporal graph neural networks (TGNNs) that mostly influence the prediction for a better ability of explanation. It employs an generative approach including the steps, motif extraction and sampling, to retrieve the structures with explainability. Experiments show that the proposed TempMe improves the state-of-the-art explainable GNNs and TGNNs in prediction.


**Strengths:**

1. Explainable AI in motif exploration is a promising research field that has not been widely studied.
2. This paper is written in good organization. The motivation and the targeted problem are illustrated clearly.
3. The designed procedure of method is clearly stated as well.


**Weaknesses:**

1. The novelty seems not sufficient as claimed.
2. More related research should be compared in the section of related work.
3. The experimental evidence is not sufficient to support the proposed approaches to address the claimed issues in prior research.

Please see the detailed comments in the review block - Questions.


**Questions:**

1. The novelty seems not sufficient. It is claimed as the first work that utilizes temporal motifs for explanation. However, the compared state-of-the-art [23] targeted finding out sub-graph structures of TGNNs for explainability enhancement as well. Therefore, the claimed novelty is questionable.
2. More related research should be compared in the section of related work. Following question 1,  it is strongly suggested to state the major differences of targeted research problems from the previous work [23]. In addition, as described in Lines 47-53, there are prior studies employing temporal motifs underlying generative mechanisms as well as the proposed work. The differences of contributions are not clearly illustrated in the manuscript though. Hence, the research value of this paper is inadequate accordingly.
3. The experimental evidence is not sufficient to support the proposed approaches to address the claimed issues in prior research. As stated in Lines 34-36, the existing research [20, 21] has not thoroughly studied the characteristics of temporal graphs, the consequent influence on the prediction and explanation is not clearly stated. Moreover, the research is not compared in the experiments. That is, it lacks experimental evidence to support the improvement under consideration of the temporal information over these works.
4. What is the time complexity of TempMe compared with [13, 14, 23]? Are there any trade-offs between the computational costs and the exploration of key motifs under TempMe?


**Limitations:**

1. The proposed approach is validated on the graphs with sparsity levels between 0 and 0.3, i.e., dense graphs with richer information. How about the effectiveness on the graphs with higher sparsity?

---

> ### Author Rebuttal · Authors · 2023-08-10
>
> We sincerely appreciate the valuable suggestions and positive comments on the motivation, and organization in this work. However, there are some misunderstandings in the novelty and experimental setting of this work. We would like to clarify as follows.
>
> >The novelty seems not sufficient.
>
> Our main novelty lies in the first attempt to utilize temporal motifs as the building blocks to analyze and explain the functionality of current temporal GNNs, which have never been explored before. Furthermore, we innovatively resort to the information bottleneck principle in the temporal explanation scenario. As mentioned in Line 42-46, the sub-graph structures extracted by [23] do not provide any motif-level insight and also entail high computational costs (Table 3). We believe the methodological and theoretical novelty of this paper is substantial and solid.
>
> >More related research should be compared in the section of related work.
>
> In this work, we focus on the same research problem with [23], i.e., the explainability of temporal GNNs.  Compared with [23], we are the first to utilize temporal motifs in this field. The temporal explanations generated by TempME provide knowledge of the combined effect of event sets and motif-level insight.
> In lines 47-53, we introduce how temporal motifs are used to analyze the underlying generative mechanisms in real-world systems as traditional methods. However, they have never been used to explain temporal GNNs. These previous works inspire us to utilize temporal motifs as more plausible and reliable composition units in explanation tasks. We have clearly stated our contributions in Line 77-81.
>
> >The experimental evidence is not sufficient to support the proposed approaches to address the claimed issues in prior research. Why not compare references [20,21] in the experiments?
>
> Existing work [20] mainly focuses on extracting a query-relevant subgraph for temporal knowledge graph reasoning. Though it explains the reasoning logic in temporal knowledge graphs to some extent, it requires a key “predicate” to infer a subgraph, which is specific to the knowledge graph setting. Previous work [21] is based on a sequence of static snapshots of a temporal graph, which is a coarse approximation to real-world temporal graphs with continuous time-stamped events. [20] and [21] either cannot process complicated dependencies between massive interactions in real-world scenarios or cannot be applied to explain general temporal GNNs under our problem formulation. We didn’t involve [20,21] in our baselines since it might be problematic to compare models under different problem settings. We believe the extensive experiments (in terms of accuracy, sparsity, connectivity, efficiency, etc.) and outstanding results over state-of-the-art baselines to be sufficient to support the superiority of TempME.
>
> >What is the time complexity of TempME compared with [13,14,23]? Are there any tradeoffs between computational cost and the key motifs exploration?
>
> The time complexity of TempME is discussed in Appendix D.3. For a thorough review, our TempME and PGExplainer[14] are more efficient than GNNExplainer[13] and TGNNExplainer[23]. TempME and PGExplainer learn a neural network to predict the importance score of a certain edge / motif, which is shared by all edges in the given graph. However, GNNExplainer and TGNNExplainer require retraining / re-searching individually for each given instance. To infer an explanation for a given instance, the time complexity of GNNExplainer is $\mathcal{O}(T|E|)$, where $|E|$ is the number of edges in the computational graph, $T$ is the number of retraining epochs. The time complexity of PGExplainer is $\mathcal{O}(|E|)$. However, these explainers ignore time information and fail to capture complicated interaction dependencies.
>
> TGNNExplainer is an MCTS-based method, where the search space increases exponentially w.r.t tree depth. The time complexity of TGNNExplainer with navigator acceleration is $\mathcal{O}(NDC)$, where $N$ is the number of rollouts, $D$ is the expansion depth of each rollout and $C$ is a constant including inference time of navigator and other operations. The inference time of TempME is mainly determined by the sampling process, which results in a complexity of $\mathcal{O}(Cl)$ where $C$ is the number of motifs and $l$ is the maximum length of the motif. Empirical results of the runtime are given in Table 3.
>
> We also conduct ablation studies on computational cost, motif exploration and model performance in Figure 4(a) and Figure 6 in Appendix E.3. By setting appropriate $C$ and $l$, TempME archives the best tradeoff between performance and efficiency.
>
> >Clarification to “The proposed approach is validated on the graphs with sparsity levels between 0 and 0.3. What about the graphs with higher sparsity?”
>
> Here are some misunderstandings. We would like to clarify that the proposed approach is validated on graphs with any level of sparsity. The sparsity levels between 0 and 0.3 is used to control the size of the explanatory subgraph w.r.t. the original graph size, since the explanation accuracy is sensitive to the explanation size. We actually conduct experiments on datasets with a wide range of sparsity levels. We also report the dataset statistics in Table 6 in Appendix E.1 for reference.

---

> > ### Comment · Reviewer_M5gq · 2023-08-14
> >
> > Thanks for the detailed response. Most of the issues are addressed. However, there are some some points should be clarified.
> >
> > 1. Following the response to Q2, it is suggested to clearly state that the studies [24-31] verify the important role of temporal motifs by what kinds of analyses or observation. As a meanwhile, it is strongly encouraged to highlight that this work is developed based on those previous verification results. Otherwise, it is confusing if these prior works utilize the temporal motifs for the enhancement of the explainability of TGNNs as well.
> >
> > 2. For the response to Q3, I understand the focused problems are subtly different. What I am looking forward is to see the improvements after the TGNNs are incorporated with the proposed temporal information. In the personal view, it is essential to compare the explainability of the graphs w/. and w/o the motif-level insights based on the evidence of either experimental or theoretical results. Otherwise, it may be hardly to be convinced that the characteristics of the "found motifs" under the proposed approach has a substantial enhancement of the explainability over those previous approaches without incorporating the temporal information as claimed in the section of related works.

---

> > > ### Author Response · Authors · 2023-08-14
> > > **Response to Reviewer M5gq**
> > >
> > > We appreciate your time and effort spent! We will definitely adopt your suggestions to further improve paper writing on previous works and intrinsic insights in our revised version. To address your concern and ensure a clearer understanding, we'd like to clarify below.
> > >
> > > 1. Previous works [24~31] have successfully demonstrated the pivotal role of temporal motifs in unraveling the complexity and dynamics of real-world networks. These traditional methods typically investigate the frequency and statistical distribution of temporal motifs to gain insights into the behavior and evolution of real-world networks. For example, in biological networks, the temporal motifs could help in pinpointing critical events in biochemical reactions. In financial networks, the occurrence of specific triadic motifs is a strong indicator of impending financial crisis, offering predictive insights. Inspired by these real-world applications of temporal motifs, we harnessed the potential of temporal motifs to uncover the decision logic of temporal GNNs, thereby improving their transparency and explainability.
> > >
> > > 2. Compared with previous methods w/o temporal motifs, the enhancements achieved through the incorporation of temporal motifs are mainly three-fold.
> > >     - **Accuracy and Fidelity Enhancement:** By integrating temporal motifs, we have observed an explainability improvement. First, the accuracy of generated explanations sees a notable increase. This is substantiated by previous research that demonstrates how temporal motifs can effectively model the evolution of dynamic networks. Moreover, the motif-level information bottleneck principle serves as a foundation, ensuring the generated explanations exhibit both maximized fidelity and minimized sparsity. The empirical evidence presented in *Table 1* and *Figure 3* affirms this positive impact on explanation accuracy and fidelity.
> > >     - **Human-Intelligible Explanations:** The introduction of temporal motifs imparts a unique property of "cohesiveness" to our explanations. This attribute arises naturally due to the inherent nature of temporal motifs. Through the extraction of interaction-related motifs, the explanations transcend singular events and provide an encompassing view of motif importance. Empirical results on the “cohesive” property are shown in *Table 2*. Visualizations of the explanations w/. motif and w/o motif are provided in our general response (please refer to the *1-page pdf*).
> > >     - **Link Prediction Performance Enhancement:** The benefits of incorporating motif-level insights extend beyond explainability improvements. It also positively influences the link prediction performance of TGNNs. This enhancement is attributed to the appropriate motif encoding that effectively captures both the spatial and temporal roles of each event. The empirical results in *Table 6* and *Table 9* substantiate the consistent link prediction enhancement across various message-passing-based TGNNs, such as TGAT, TGN, and GraphMixer.
> > >
> > > Thanks again for your suggestions. ​​We hope that the above clarification improves your confidence in our work. Let us know if you have any further questions/concerns.

---

> > > > ### Comment · Reviewer_M5gq · 2023-08-17
> > > >
> > > > According to the provided strengths and improvements on TGNNs, I raise my rating to borderline accept.

---

### Official Review · Reviewer_uwpb · 2023-07-31

**Soundness:** 3 good
**Presentation:** 3 good
**Contribution:** 3 good
**Rating:** 6
**Confidence:** 3

**Summary:**

The Explainability of TGNN is essential for human understanding of model prediction results. This paper proposed a framework, TempME, to construct meaningful, cohesive explanations by utilizing temporal motifs. The framework consists of temporal motif extraction, motif encoder, information-bottleneck-based importance score generator, and an explanation process. Benefiting from the information bottleneck principle, TempME can balance the explanation accuracy and compression. Experiments on six real-world temporal graph datasets were conducted with three TGNN backbones. The result shows that the selected temporal motifs not only reason the predictions of the backbones but also enhance their link prediction accuracy.

**Strengths:**

-- Connecting the information bottleneck principle with the temporal motif explanation task is considered novel.

-- The experiment results are promising. The TempME achieves up to an 8.21% increase in explanation accuracy across real-world datasets. Besides, up to 22.96% improvement in the performance of TGNNs shows the effectiveness of discovered temporal motifs.

-- The author provides comprehensive theoretical analysis and extensive experiments.

-- The TempME is efficient in terms of the inference time for producing explanations.

-- The paper is well-organized and easy to follow.


**Weaknesses:**

-- The performance of the tempME might be susceptible to temporal motif extractions in the first stage, but fewer discussions on other motif mining algorithms.

-- The proposed method assumed temporal motifs can capture the dynamics of the graph, which might not hold for real-world scenarios. For instance, external events or user preferences can also affect the explanations.

-- The ACC-AUC of TempME in Table 1. outperforms baselines in most cases. However, the performance improvements of the Enron dataset are not significant. The reviewer also finds the results of the Enron dataset in Table 4. show less improvement in the performance of TGNN models with motif embedding. It would be better to investigate the limitation of the proposed method.


**Questions:**

As stated in the weakness.

**Limitations:**

The proposed method assumed temporal motifs can capture the dynamics of the graph, which might not hold for real-world scenarios. For instance, external events or user preferences can also affect the explanations.

---

> ### Author Rebuttal · Authors · 2023-08-10
>
> Thanks for your valuable feedback and positive comments on the novelty, experimental results, theoretical analysis, and paper writing in this work. We address below the potential concerns.
>
> >The performance of TempME might be susceptible to temporal motif extraction in the first stage. Discussions on other motif mining algorithms.
>
> Thanks for bringing this up. Exact motif counting results in high memory and large computation complexity. Therefore, most other motif extraction algorithms (e.g. MAVisto, Kavosh) mainly aim to reduce approximation error of motif counting with less memory and CPU time. However, in TempME, the target of motif extraction in the first stage is to collect a candidate set of expressive temporal motifs, instead of approximating the exact motif counting. Therefore, the performance of TempME is not sensitive to the temporal motif extraction algorithms. We will add more discussions on other motif mining algorithms in our revised version.
>
> >Clarification to “The proposed method assumed temporal motifs can capture the dynamics of the graph, which might not hold for real-world scenarios”
>
> We totally agree that dynamic systems can be influenced by a multitude of external factors or emergent behaviors. However, many works have demonstrated that temporal motifs are essential building blocks of complex dynamic systems in real-world scenarios [1,2], which help us to gain a deeper understanding of the system’s functioning. We believe that this manuscript has a unique value in investigating how temporal motifs facilitate explainability. It is also a necessary first step before one can further extend to a multitude of factors in future works.
>
> [1]Temporal motifs in time-dependent networks
>
> [2] Motifs in Temporal Networks
>
> >The performance improvements of the Enron dataset are not significant compared with other datasets in Table 1 and Table 4.
>
> Thanks for the interesting question. We also noticed that the performance improvement on Enron is not as significant as other datasets. One possible reason is that temporal interactions are the most dense in Enron, compared with other datasets in experiments. We refer to Table 6 in Appendix E.1 for dataset information. Here we define the interaction density as the ratio of #links to #nodes (regardless of time durations). It might be more challenging to analyze temporal graphs with high interaction density with motifs, since there could be complex interactions and mixed effects of multiple motifs. We are more than happy to further analyze and discuss potential limitations in our final version.

---

> > ### Comment · Reviewer_uwpb · 2023-08-13
> >
> > Thank you for your clarifications. I am satisfied with the explanations for the first two concerns. However, I am not sure whether the limited performance improvements in the Enron dataset can be simply attributed to dense temporal interactions. In Table 4, US Legis also has a higher interaction density compared to Wikipedia, UCI, and Reddit, but the improvement is significant. The statement “It might be more challenging to analyze temporal graphs with high interaction density with motifs…” would be more convincing if more evidence were provided.

---

> > > ### Author Response · Authors · 2023-08-14
> > > **Response to Reviewer uwpb**
> > >
> > > Thanks for keeping up the discussion! Your question has motivated us to further investigate the characteristics of the Enron dataset compared with other datasets used in our experiments.
> > >
> > > Enron is an email network where interaction events are emails exchanged among employees. After more statistical studies, we observed that there are multiple identical interactions in the Enron dataset. For example, the interaction between Node115 and Node124 at timestamp 533820 recurs four times. On average, each distinct interaction is accompanied by **3.284** exactly identical interactions within this dataset. (Refer to the following table for a comparison of the average occurrences of identical interactions across all datasets)
> > >
> > > | UCI   | USLegis | Wikipedia | Reddit | Canparl | Enron     |
> > > |-------|---------|-----------|--------|---------|-----------|
> > > | 1.001 | 1.007   | 1.000     | 1.000  | 1.000   | **3.284** |
> > >
> > > While this phenomenon might be deemed reasonable within the context of email networks, wherein multiple emails are dispatched to the same recipient at identical timestamps, it is not a common phenomenon in other datasets and many real-world scenarios. For consistency with existing literature [1,2] we restrict the timestamp of the next sampled event strictly earlier than the previous event, which is also a necessary condition of underlying causality between interactions [1]. Consequently, many identical interactions in the Enron dataset are not sampled within a temporal motif, thereby potentially degrading the performance improvement of TempME over this specific dataset.
> > >
> > > It is crucial to emphasize that **our proposed algorithm adeptly handles scenarios involving multiple interactions at the same timestamp between different node pairs, as well as multiple interactions between the same node pair but at different timestamps.** However, we restate the limitations we articulated earlier: “One limitation of TempME is analyzing temporal graphs characterized by **high interaction density between the same node pairs at the same timestamp**, such as Enron dataset”.
> > >
> > > We will add more discussion on limitations. Again, thanks for your attention and discussion.
> > >
> > > [1]Temporal motifs in time-dependent networks
> > >
> > > [2]Motifs in Temporal Networks

---

> > > > ### Comment · Reviewer_uwpb · 2023-08-15
> > > >
> > > > Thank you so much for the further investigation. The explanations make sense to me.

---

### Author Rebuttal · Authors · 2023-08-10

We sincerely appreciate the reviewers for their valuable time and effort in reviewing this manuscript. We have extended our experiments, which we detail below. We appreciate your feedback on this, and we agree that additional empirical verification would better support our proposed framework and thus make the paper stronger.

We provide a visualization of real-world explanation examples on Wikipedia. The base model is TGAT with two layers. We demonstrate the explanations generated by Grad-CAM, TGNNExplainer, and the proposed TempME. We can make the following observations.
- Compared with TGNNExplainer, Grad-CAM tends to generate more "cohesive" explanations. One possible reason is that topologically close events tend to have close gradients. In contrast, the explanation generated by TGNNExplainer contains more isolated events, which degrades the inspiration that explanations could bring us.

- Compared with Grad-CAM and TGNNExplainer, the proposed TempME not only can generate more "cohesive" explanations but also provide additional knowledge on the combined effect of explanatory events from a unique perspective of temporal motifs. TempME is capable of capturing the importance of each motif to the dynamic of the temporal graph.

According to the suggestions from the reviewer, we add $CF^2$[1] as a baseline, which is a state-of-the-art explainer on static graphs. We report our preliminary results on Wikipedia and UCI in Table 1. Complete results will be given in the final version.

We hope that the experimental support improves your confidence in our work. Again we thank you for your feedback and we continue to look forward to the coming discussion.

[1]Tan, Juntao, et al. "Learning and evaluating graph neural network explanations based on counterfactual and factual reasoning." Proceedings of the ACM Web Conference 2022. 2022.

---

### Decision · Program_Chairs · 2023-09-21

**Decision:**

Accept (poster)

**Comment:**

The reviewers raised a few concerns, including the assumption, the performance studies, etc., on this paper. The authors addressed most of the concerns during the rebuttal/discussion phase, and the reviewers were finally all positive about the work. After reading the reviews and discussions, and also taking the scores into account, I am recommending this paper be accepted.